# Tia1 dependent regulation of mRNA subcellular location and translation controls p53 expression in B cells

Manuel D. Díaz-Muñoz [1,2], Vladimir Yu. Kiselev [3,7], Nicolas Le Novère [3], Tomaz Curk [4], Jernej Ule [5,6] & Martin Turner [1]

Post-transcriptional regulation of cellular mRNA is essential for protein synthesis. Here we describe the importance of mRNA translational repression and mRNA subcellular location for protein expression during B lymphocyte activation and the DNA damage response. Cytoplasmic RNA granules are formed upon cell activation with mitogens, including stress granules that contain the RNA binding protein Tia1. Tia1 binds to a subset of transcripts involved in cell stress, including p53 mRNA, and controls translational silencing and RNA granule localization. DNA damage promotes mRNA relocation and translation in part due to dissociation of Tia1 from its mRNA targets. Upon DNA damage, p53 mRNA is released from stress granules and associates with polyribosomes to increase protein synthesis in a CAP-independent manner. Global analysis of cellular mRNA abundance and translation indicates that this is an extended ATM-dependent mechanism to increase protein expression of key modulators of the DNA damage response.

[1] Laboratory of Lymphocyte Signalling and Development, The Babraham Institute, Cambridge CB22 3AT, UK. [2] Centre de Physiopathologie Toulouse-Purpan, INSERM UMR1043 / CNRS U5282, Toulouse 31300, France. [3] Laboratory of Signalling, The Babraham Institute, Cambridge CB22 3AT, UK. [4] University of Ljubljana, Faculty of Computer and Information Science, Ljubljana, Slovenia. [5] Department of Molecular Neuroscience, UCL Institute of Neurology, Queen Square, London WC1N 3BG, UK. [6] The Crick Institute, 1 Midland Road, London NW1 1AT, UK. [7] Present address: Wellcome Trust Sanger Institute, Wellcome Trust Genome Campus, Cambridge CB10 1SA, UK. Correspondence and requests for materials should be addressed to M.D.D.-M. (email: manuel.diaz-munoz@babraham.ac.uk or manuel.diaz-munoz@kcl.ac.uk)

Programmed DNA damage occurs during B-cell development to generate highly diverse immunoglobulins (Ig). In pro- and pre-B cells, the formation of double strand DNA breaks (DSB) is required for recombination of the variable (V), joining (J), and diversity (D) gene segments of the Ig loci (VDJ recombination) to generate a functional B cell receptor (BCR)[1]. Cytosine deamination by activation-induced cytidine deaminase (AID) in mature B cells allows class switch recombination (CSR) and somatic hypermutation (SHM), two mechanisms that increase the antibody repertoire upon antigen encounter[2–4]. B lymphocytes rely on constant monitoring of genome integrity. DNA damage repair (DDR) pathways, including homologous recombination (HR), non-homologous end joining (NHEJ), base excision repair (BER) and mismatch-mediated repair (MMR), are finely coupled to cell cycle progression[5], differentiation[6] and apoptosis upon B-cell activation to prevent B cell tumour transformation[7]. Cell cycle checkpoints are essential for timely DNA repair. ATM and p53 activation enforce both G1 and G2 cell cycle arrest and activation of DDR pathways[8, 9]. ATM[−/−] and p53[−/−] B cells show defects in VDJ and class-switch recombination[10–12]. Notably, mice deficient in p53 and NHEJ or H2A.X develop aggressive B-cell lymphomas[13–15]. Lack of VDJ and class-switch recombination in the absence of NHEJ repair is not rescued by p53 deficiency[13], which highlights the role of p53-mediated apoptosis in preventing the survival and expansion of tumour-transformed B lymphocytes.

P53 expression and activity is regulated both at the level of mRNA and protein[16–18]. It has been proposed that Bcl6 inhibition of p53 transcription is required for promoting error-prone DNA repair in germinal center (GC) B cells undergoing clonal expansion, CSR and SHM without inducing an apoptotic response[19]. However, recent characterization of the transcriptomes of follicular and GC B cells by deep sequencing indicates that p53 mRNA abundance does not change substantially[20, 21], suggesting that other mechanisms in addition to transcription are important for p53 expression in B lymphocytes. Here we describe a general post-transcriptional mechanism that uncouples mRNA expression and protein synthesis upon B-cell activation. p53 protein is hardly detected in activated B lymphocytes, at least in part due to localization of its mRNA within cytoplasmic RNA granules where translation into protein is inhibited.

Cytoplasmic RNA granules are key modulators of post-transcriptional gene expression[22]. They are microscopically visible aggregates of ribonucleoprotein (RNP) complexes often formed upon stress-induced translational silencing. Disassembly of polyribosomes from messenger RNA can drive the formation of two RNA granule types in mammalian cells with distinct protein composition and functions: processing bodies (PBs) contain components of the mRNA decay machinery[23, 24]; and stress granules (SGs) contain members of the translational initiation complex[25, 26] and several translational silencers, including Tia1 and Tia-like 1 (Tial1), that contribute to polysome disassembly and mRNA translational arrest. Although stress-induced PBs and SGs have been extensively studied in model cell systems, very little is known about whether they are formed and functional in primary cells. Here, we present evidence that formation of RNA granules controls post-transcriptional gene expression upon B cell activation. Exchange of mRNA transcripts between SGs and polysomes allows rapid translation of key modulators of the DNA damage response.

The RNA-binding protein Tia1 has an important role in SG nucleation. Tia1 overexpression induces the assembly of SGs in the absence of stress[25], whereas depletion of the glutamine-rich prion-related domain of Tia1 impairs SGs formation[27]. Tia1 and Tial1 are essential for cell development and differentiation[28, 29].

Tial1 knockout (KO) mice are embryonic lethal, whereas 50% of Tia1-KO mice die by 3 weeks of age. Tia1-KO mouse survivors have profound immunological defects associated with increased production of TNF and IL-6[29]. By using individual-nucleotide resolution UV crosslinking and immunoprecipitation (iCLIP)[30] and nucleus-depleted cell extracts we have identified the mRNA targets of Tia1 in activated B lymphocytes. Tia1 protein accumulates in SGs and is associated with translationally silenced mRNAs including that encoding the transcription factor p53. Genome-wide analysis of mRNA abundance and translation highlights the importance of mRNA subcellular location and translational repression for B-cell activation and clonal expansion. DNA damage induces Tia1 dissociation from its mRNA targets and translocation of these mRNAs out of SGs. This enables rapid protein synthesis of key transcription factors for cell cycle arrest, DNA damage repair and apoptosis.

## Results

**RNA granules are assembled upon B-cell activation**. Upon activation with antigens, resting B lymphocytes become metabolically active and initiate a genetic program for cell growth, division and differentiation. Analysis by RT-qPCR showed that the abundance of transcripts encoding protein components of SG and PB increased after cell treatment with the mitogens LPS and antiCD40 + IL4 + IL5 (Fig. 1a and Supplementary Fig. 1A) indicating that post-transcriptional control of mRNA processing and storage might increase upon B-cell activation. Indeed, protein expression of Dcp1a and Tia1, two specific markers for PB and SG, respectively, was increased in B cells stimulated with mitogens (Fig. 1b and Supplementary Fig. 1B). PBs and SGs were rare in ex vivo isolated B cells but accumulated upon B-cell activation (Fig. 1c, Supplementary Fig. 1C, D). Co-staining of Dcp1a and Tia1 in activated B cells (Supplementary Fig. 1E) suggested that PBs and SGs were discrete entities, although they could be found located proximally to each other (Supplementary Fig. 1E) as previously seen in non-lymphoid cells[31]. The eukaryotic initiation factor eIF3η co-localized with Tia1 (Supplementary Fig. 1F) indicating that translation initiation factors accumulate in SGs as previously shown[26].

In order to identify candidate mRNA transcripts that could be silenced within SGs, we performed iCLIP for Tia1 using cytoplasmic extracts of activated B lymphocytes (Fig. 1d and Supplementary Fig. 2A). Analysis of genomic mappability of the cDNAs uniquely identified by iCLIP showed that Tia1 preferentially binds to U-rich elements in the 3′UTR of mature transcripts (Supplementary Fig. 2B, C). Fewer than 5% of the reads were mapped to introns (Fig. 1d and Supplementary Fig. 2C). This result suggested that depletion of the cell nucleus was highly efficient. Peak enrichment analysis identified mRNA transcripts of 3888 genes whose 3′UTR was targeted by Tia1 in the two independent experiments performed (Fig. 1e and Supplementary Data 1). Gene set enrichment analysis showed that Tia1 binds RNAs associated with the signalling and metabolic switch occurring upon B-cell activation and genes regulating cell growth and apoptosis (Fig. 1f and Supplementary Fig. 2D). Although it is possible that not all transcripts bound by Tia1 will be located in SGs, we hypothesized that they are likely to be under post-transcriptional silencing to a certain extent as Tia1 and other SG-components like G3BP1 are not associated with ribosomes in B cells (Supplementary Fig. 3), which is consistent with previous reports[26, 32].

**DNA damage induces global changes in mRNA translation**. AID expression upon B-cell activation induces DNA double strand breaks and triggers CSR and SHM[2, 6]. To evaluate global

mRNA translation in activated B cells undergoing a DNA damage response we used etoposide, a topoisomerase II inhibitor that induces DNA double-strand breaks. Using sucrose gradients, we separated mature mRNA transcripts into 13 fractions based on their ribosomal density. Quantitation of mRNA transcripts within monosomes, light polysomes and heavy polysomes was performed by 3′end RNA sequencing (3′ RNA-seq). Cellular mRNA abundance was quantified by mRNAseq (Supplementary Data 2). Integration of the results from both methodologies allowed us to identify global changes in mRNA abundance, mRNA association with ribosomes (hereafter referred to as ribosome association) and mRNA distribution along polysome

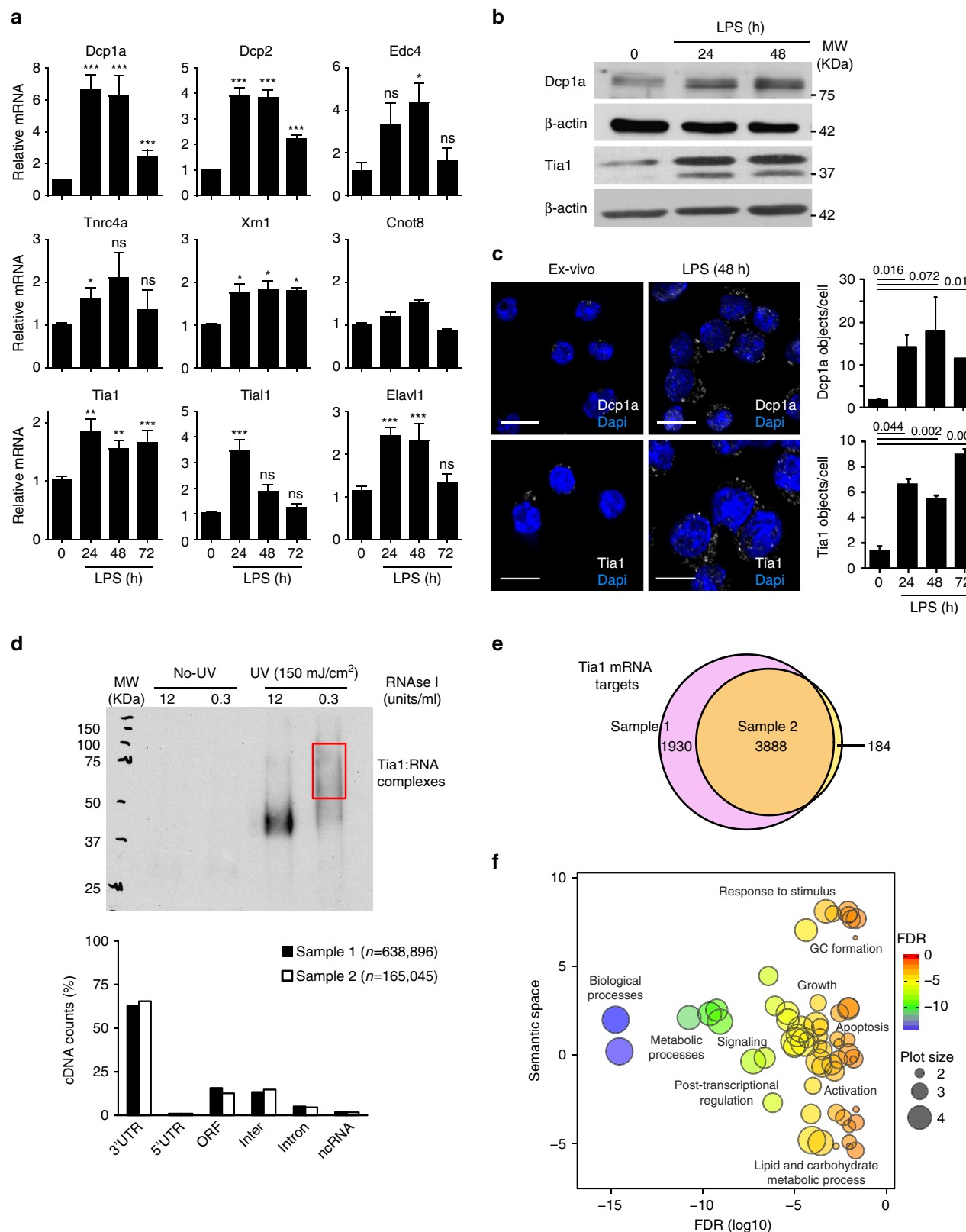

fractions (hereafter referred to as polysome distribution; Fig. 2a). Cellular mRNA abundance of transcripts encoded by 6340 genes (Ensembl gene annotation) changed upon induction of DNA damage with etoposide (Fig. 2b and Supplementary Data 2). Although etoposide globally reduced polysome assembly in B cells (Supplementary Fig. 3A), 53% of these genes (3355 out of 6340 genes) showed no significant changes in the amount of mRNA associated with ribosomes or their distribution along polysomes (Fig. 2b, *black dots*), suggesting that changes in mRNA abundance are not always linked to changes in translation.

A total of 3580 genes were differentially translated upon DNA damage induction (Fig. 2b and Supplementary Data 3), with changes in the polysome distribution mostly associated with changes in the monosome and heavy fractions (Fig. 2c and Supplementary Data 3, 4). 83% of these genes (2985 out of 3580 genes) were differentially expressed, linking their cellular mRNA abundance with translation (Fig. 2b). In all, 595 genes showed significant changes in ribosome association but not in cellular mRNA abundance, indicating a prominent role for translational control (Fig. 2b, *red dots*). Differential ribosome association highly correlated with changes in polysome distribution (Fig. 2d, e). Twenty percent of the 3355 differentially expressed genes that showed no changes in the amount of RNA associated with ribosomes, showed a change in polysome distribution as measured by a significant change in mRNA abundance in any one of the 13 polysome fractions. By contrast, over 50% of the 3580 genes that showed global changes in ribosome association showed significant changes in the amount of mRNA in at least one of these fractions in our polysome distribution analysis (Fig. 2d). Analysis of significant changes in mRNA abundance in monosome, light and heavy polysome fractions showed that mRNAs with differential mRNA expression and ribosome association were mostly found in the heavy polysome fractions (Fig. 2d). Similarly, over 60% of mRNAs that showed changes in polysome distribution were differentially expressed or associated with ribosomes (Fig. 2e). Less than 10% of mRNAs showed changes in "Riboload", a measure of mRNA distribution in heavy polysomes that is independent of any changes in mRNA abundance (Fig. 2e). This suggests that changes in polysome distribution caused by etoposide are linked to both changes in mRNA abundance and/or ribosome association. These proportions did not change if mRNAs were classified based on changes in their abundance in the monosome, light or heavy polysome fractions (Fig. 2e). Taken together, these results show that a significant fraction of genes in B cells are subject to post-transcriptional regulation of mRNA translation following DNA damage.

**Tia1 regulates mRNA translation upon etoposide treatment.** Tia1 mRNA targets in B cells were highly responsive to DNA damage. Total mRNA abundance and/or the amount of mRNA associated with ribosomes changed for 77% of Tia1 mRNA targets following treatment with etoposide (Fig. 3a). The greater the number of unique iCLIP cDNAs mapped to a given Tia1 mRNA target the greater the likelihood for changes in mRNA polysome distribution (Fig. 3b). Abundance and translation of most Tia1 mRNA targets was reduced after DNA damage induction (Fig. 3c). These were genes related to B-cell metabolism and proliferation (Fig. 3d and Supplementary Data 5) and they resembled the overall reduction in mRNA abundance and translation of highly expressed genes (Fig. 4a, b).

Notably, the translation of 20% of Tia1- mRNA targets was increased upon etoposide treatment including those related to the cell stress response (Fig. 3c, d and Supplementary Data 5). Analysis of mRNA distribution along heavy polysomes indicated that changes in mRNA translation were largely linked to differential mRNA abundance, independently of whether they were targets of Tia1 (Fig. 4c). Among the 5% of Tia1 mRNA targets with differential mRNA translation, but no changes in cellular mRNA abundance, we identified mRNAs encoding two of the key transcription factors that control the DNA damage response, p53 and Hif1α (Fig. 4d). Thus Tia1 may regulate the DNA damage response in etoposide treated B cells.

**Tia1 controls p53 mRNA translation.** Following B-cell activation in vitro p53 mRNA abundance increases 1.5-fold (Supplementary Fig. 4A, B), although p53 protein is hardly detectable at any time point (Supplementary Fig. 4C). Total p53 mRNA levels remained constant after B-cell treatment with etoposide (Fig. 4e and Supplementary Fig. 4D). By contrast, p53 mRNA association with ribosomes increased, suggesting that a higher number of p53 transcripts were used for protein synthesis after induction of a DNA damage response (Fig. 4e). Indeed, distribution analysis of p53 transcripts in the different polysome fractions showed a significant increase in the number and proportion of transcripts in heavy polysomes upon B cell treatment with etoposide indicating greater p53 mRNA translation (Fig. 4f, Supplementary Fig. 4D and Supplementary Data 6). Production of active p53 protein after DNA damage induction was reflected by an increase in total mRNA abundance and polysome association of mRNAs encoding Gadd45a and Cdkn1a (p21; Supplementary Fig. 4D and Supplementary Data 6), two previously described p53-dependent genes[33, 34] the expression of which was partially reduced when B cells were pre-treated with the p53 inhibitor Pifithrin-α (Supplementary Fig. 4E, F). In summary, our data suggest that post-

**Fig. 1** Stress granules and P-bodies are assembled after B-cell activation. **a** Quantitation by RT-qPCR of mRNA expression of PBs and SGs components in B cells after LPS activation. Data from at least two independent experiments are shown as mean + s.e.m. (Mann–Whitney test, *ns* non-significant, * $P < 0.05$, ** $P < 0.01$, *** $P < 0.001$). **b** Immunoblot analysis of Dcp1a and Tia1 protein abundance in LPS- activated B cells. **c** Visualization of Dcp1a and Tia1 containing granules in ex vivo and LPS-activated B cells by confocal microscopy. Nuclei were stained using Dapi. The number of Dcp1a and Tia1- containing granules per cell is shown in the *right panels*. Quantitation was performed using Volocity Image software based on fluorescence accumulation within granules. A minimum of six confocal images from three independent experiments were analyzed. Data shown as mean + s.d. Statistical analysis of the number of objects in treated B cells compared to untreated B was performed using a Mann–Whitney test. (*Scale bar* = 10 μm). **d** Detection and feature distribution analysis of Tia1 binding to its mRNA targets in the cytoplasm of B cells treated with LPS for 48 h. *Top panel*, autoradiography showing Tia1:RNA complexes labelled with $^{32}$P-ATP after complete or partial RNA digestion with RNaseI. *Red box* indicates the RNA:protein complexes purified for cDNA library preparation and iCLIP analysis. *Bottom panel*, distribution analysis of iCLIP cDNA counts mapped to the indicated genomic features (3′UTR, 5′UTR, ORF, intergenomic, intron and ncRNA). Data from the two Tia1 iCLIP experiments performed using B-cell cytoplasmic extracts are shown. **e** *Venn diagram* showing the number of Tia1 mRNA targets identified in two iCLIP experiments. Only mRNA targets with at least one significantly enriched binding site in the 3′UTR were considered (transcripts collapsed by gene id, peak enrichment analysis, FDR < 0.05). **f** Gene ontology (GOrilla and REViGO) analysis of Tia1 mRNA targets in activated B cells. Genes which expression was detected in our mRNAseq libraries from LPS-activated B cells was used as a background gene list for GO enrichment analysis. Bubble plot sizes indicate the generality of the GO terms. Larger bubbles relate to general terms whereas smaller bubbles imply more specific GO terms

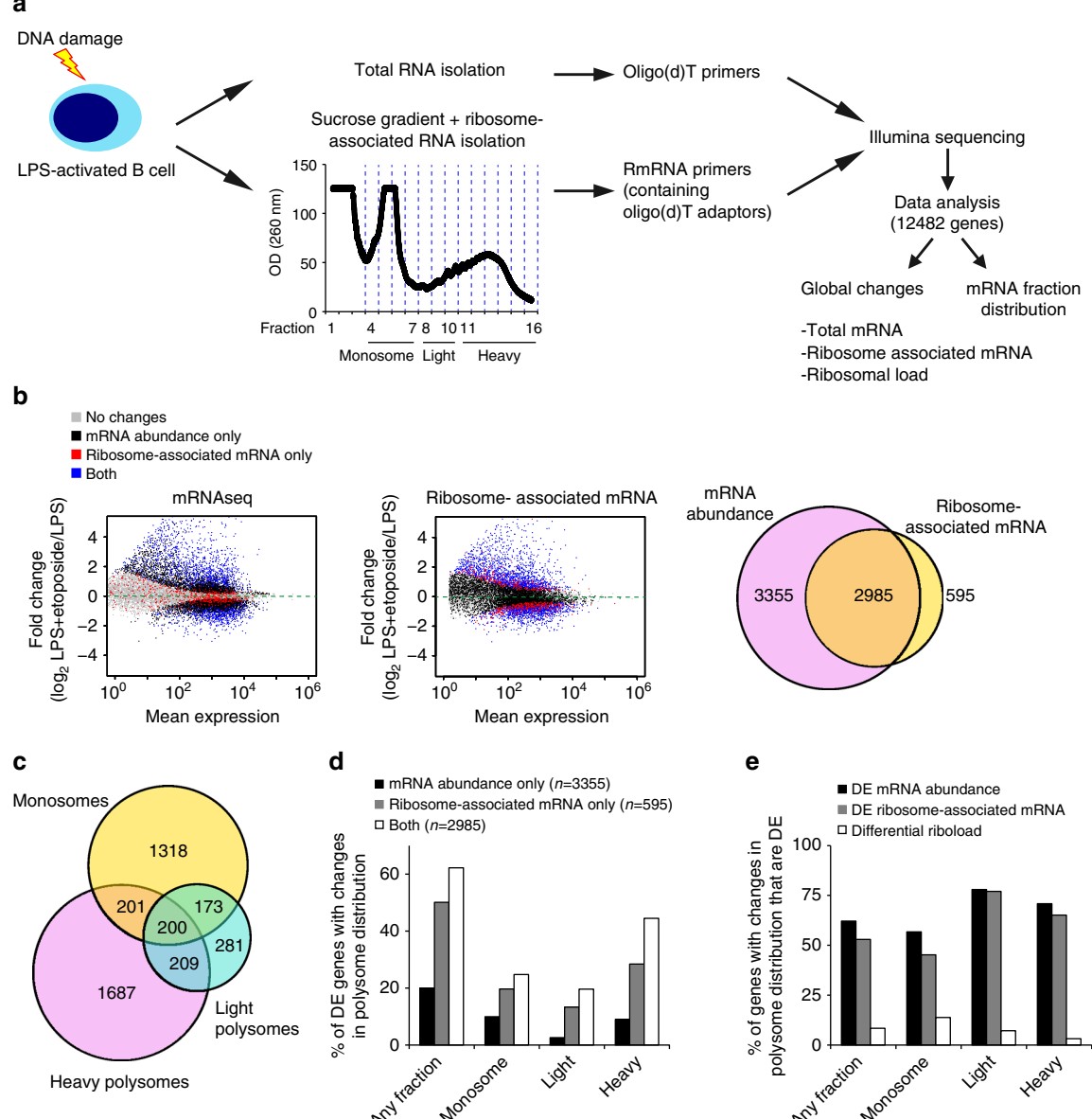

**Fig. 2** DNA damage induces changes in mRNA translation decoupled from total mRNA abundance. **a** Experimental set up for the study of etoposide- induced changes in total mRNA abundance (mRNAseq) and mRNA translation (Polyribosome profiling followed by 3'end RNA sequencing; PolyRibo-3' RNA-seq) in B cells. Libraries from four biological replicates were generated for both mRNAseq and PolyRibo-3' RNA-seq. Changes in total cellular mRNA abundance, changes in ribosome-associated mRNAs and changes in polysome distribution after induction of DNA damage were analyzed using DESeq2 (transcripts collapsed by gene id). **b** Visualization as MA plots of changes in B-cell transcriptome and B-cell translatome after etoposide treatment (20 μM, 4 h) (LPS = L, LPS + Etoposide = LE). *Top panel*, MA plot showing the mean expression of mRNA transcripts in B cells vs. changes in total mRNA abundance after treatment with etoposide (mRNAseq). *Middle panel*, MA plot showing the mean abundance of ribosome- associated mRNAs vs. changes in mRNA translation induced by etoposide (PolyRibo-3' RNA-seq) (grey—unchanged mRNAs in both datasets, *black*—mRNAs that change in cellular mRNA abundance only, *red*—mRNAs with a translational change only, *blue*—mRNAs that change at both mRNA abundance and translation). *Bottom panel*, Venn-diagram with the number of mRNAs that change in abundance, translation or both. **c** *Venn-diagram* showing the number of mRNA transcripts with significant changes in monosome, light and heavy polysome fractions (mRNA fraction distribution analysis). **d** Proportion of differentially expressed (DE) mRNAs, differentially ribosome-associated mRNAs or both that show a significant change in polysome distribution. mRNAs are grouped based on significant changes in abundance in any of the fractions (fractions 4 to 16) or in monosomes only (fractions 4 to 7), in light polysomes (fractions 8 to 10) or in heavy polysomes (fractions 11 to16). **e** Proportion of genes with a change in polysome distribution (defined change in any of the fractions, monosomes, light polysomes and heavy polysomes) that are differentially expressed (DE, change in total mRNA abundance), differentially associated with ribosomes or show a change in RiboLoad (defined as the proportion of reads in heavy polysomes divided by the total of reads mapped to a given mRNA transcript; [LPS + Etoposide vs. LPS and 0.75 < RiboLoad <1.25])

transcriptional control of p53 mRNA translation by Tia1 might be is a regulatory mechanism in activated B cells that is responsible for changes in gene expression during the DNA damage response.

**Tia1 binding to p53 3′UTR controls mRNA translation.** Tia1 has been implicated in mRNA subcellular location and translational silencing[25, 29], therefore we hypothesized that Tia1 was responsible for p53 translational control in B lymphocytes. iCLIP

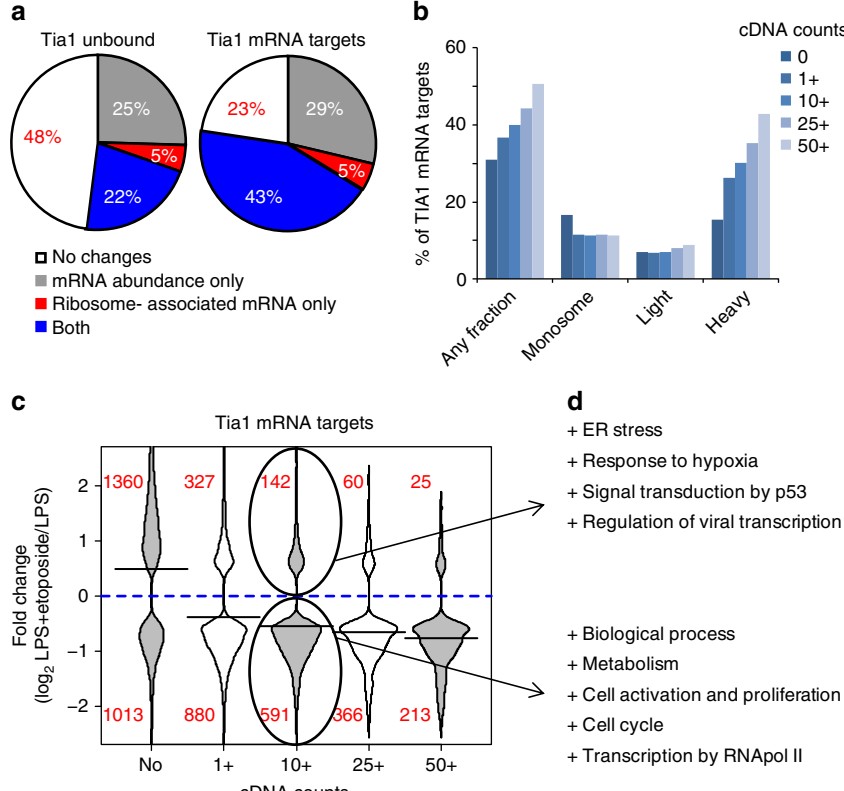

**Fig. 3** Tia1 mRNA targets are highly responsive to DNA damage. **a** Proportion of Tia1 mRNA targets compared to non-targeted mRNAs that are differentially expressed in total mRNA abundance, in ribosome-associated or both after B-cell treatment with etoposide. Tia1 mRNA targets were defined as those mRNAs in which >50 unique cDNA counts were mapped to the 3′UTR in our Tia1 iCLIP experiments. **b** Proportion of Tia1 mRNA targets with a significant change in any polysome fraction, in monosomes, light or heavy fractions. Tia1 mRNA targets were divided based on the number of unique cDNA counts mapped to the 3′UTR (1+, 10+, 25+or 50+). Tia1 untargeted mRNAs were classified as having none (0) cDNA counts mapped to the 3′UTR. **c** Distribution analysis of the fold change in ribosome-associated mRNA abundance of differentially expressed Tia1 mRNA targets compared to non-targeted mRNAs. Tia1 mRNA targets were sub-divided as in **b**. **d** Gene ontology (GOrilla and REViGO) analysis of Tia1 mRNA targets (10+ cDNA counts mapped to the 3′UTR), which mRNA translation is increased or decreased after B-cell treatment with etoposide (extended in Supplementary Data 5)

mapping of Tia1:p53 mRNA interactions showed that Tia1 binds to two U-rich sequences present in the p53 3′UTR called proximal (P) and distal (D) binding sites based on their relative proximity to p53 stop codon (Fig. 5a and Supplementary Fig. 5A). To gain mechanistic insight into how Tia1 binding to these sequences controls mRNA translation we cloned the p53 3′UTR downstream of a renilla luciferase reporter gene and used a HEK293 cell system to evaluate translation. Knock down of endogenous Tia1 increased expression of the renilla luciferase reporter gene without changes in the abundance of renilla luciferase mRNA (Fig. 5b and Supplementary Fig. 5D). The increase in luciferase activity was directly related to the extent of mRNA and protein depletion caused by each of the three different dsiRNAs used to target Tia1 (Supplementary Fig. 5B, C). No changes in activity of the luciferase reporter were observed when Hprt was knocked down (Fig. 5b and Supplementary Fig. 5B, D). Mutation of the proximal and distal Tia1 binding sites in the p53 5′UTR increased the activity of the reporter gene without changes in mRNA abundance (Fig. 5c, d and Supplementary Fig. 5E). Over expression of Tia1 in HEK293 cells reduced luciferase activity by 40% (Fig. 5d). Importantly, this reduction was abrogated by mutation of both proximal and distal Tia1 binding sites (Fig. 5d). Moreover, over-expression of a mutant Tia1 protein lacking RNA recognition motifs 1 and 2 was unable to inhibit expression of the reporter gene (Fig. 5d). Importantly, mRNA levels of the luciferase reporter remained unchanged in all

conditions tested (Supplementary Fig. 5F) indicating that Tia1 binding to p53 3′UTR controls translation without affecting mRNA abundance.

**Tia1 dissociation allows p53 expression upon DNA damage.** Analysis by confocal microscopy showed that p53 mRNA colocalized with Tia1 protein containing SG in B cells (Fig. 6a and Supplementary Fig. 6A). Tia1 protein was not detected in monosome and polysome- containing fractions by Western blotting (Supplementary Fig. 3) suggesting that p53 mRNA bound by Tia1 and contained within SGs was not quantified in our polysome fractionation and sequencing analysis. Induction of DNA damage with etoposide resulted in p53 mRNA translocation out of Tia1-containing SG (Fig. 6a and Supplementary Fig. 6A). Etoposide treatment did not affect SG number and/or size (Supplementary Figs. 1F, 6B), but promoted dissociation of Tia1 from p53 mRNA (Fig. 6b, Supplementary Fig. 6C, D). This was not due to depletion of Tia1 following DNA damage as Tia1 protein abundance in B cells was unaffected by etoposide (Fig. 6c).

DNA damage induced p53 protein expression in primary mouse B cells (Fig. 6d). To test if the p53 3′UTR could mediate the induction of p53 upon treatment with etoposide we tested the response of the 3′UTR reporter in HEK293 cells and found etoposide significantly induced the expression of the reporter (Fig. 6e) without affecting mRNA abundance when compared to

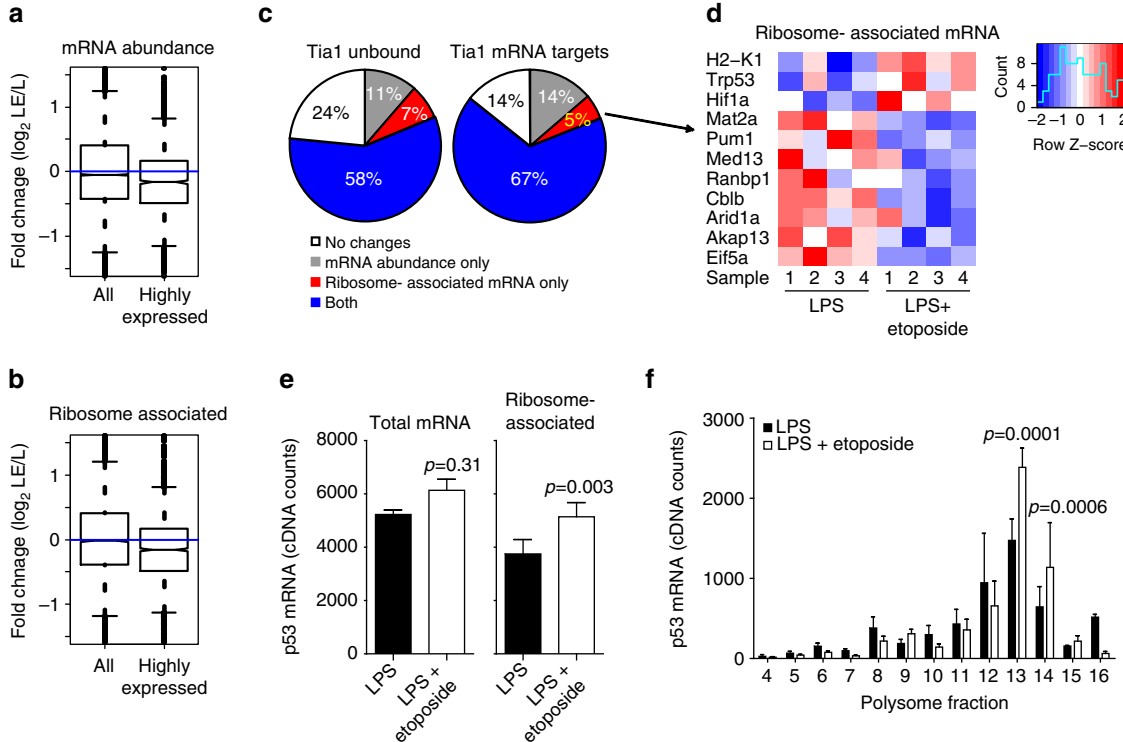

**Fig. 4** DNA damage induces mRNA translation of p53 in B cells. **a**, **b** Global analysis of changes in mRNA abundance (**a**) and mRNA association with ribosomes (**b**) after treatment with etoposide of LPS-activated B cells. Box plots were generated with data from all genes or from only a selection of highly expressed genes. These genes were selected after calculating the mean number of normalized reads annotated to each gene in our mRNAseq and in our PolyRibo-3′ RNA-seq libraries. Then, windows of expression were generated by adding or subtracting 1000 counts (mRNAseq window in log2 is 11.09 < 11.63 (mean) < 12.03; PolyRibo-3′ RNA-seq window in log2 is 9.13 < 10.61(mean) < 11.32). Fold change equal to 0 is marked by the blue line. Wilcox test was performed comparing all genes vs. highly expressed genes. In both cases $P < 2.2 \times 10^{-16}$. **c** Proportion of Tia1 mRNA targets (defined as those mRNAs with 50+ unique cDNA counts mapped to the 3′UTR in our Tia1 iCLIP experiments) and non- targeted mRNAs with a differential change in any heavy fractions that were also identified as differentially expressed at total mRNA abundance only, at ribosome-associated mRNA level or both. **d** Heatmap showing ribosome-associated mRNA abundance of Tia1 mRNA targets with a differential change in any heavy fractions and in mRNA translation. Numbers relate to each biological replicate. **e** Total cellular mRNA and ribosome-associated mRNA levels of p53 in LPS-activated B cells treated or not with etoposide quantified by mRNA-seq and PolyRibo-3′ RNA-sequencing. **f** Polysome profile of p53 mRNA from PolyRibo-3′ RNA-seq experiments. Four biological replicates were analyzed in both mRNA-seq and PolyRibo-3′ RNA-sequencing experiments. Data in **e**, **f** are shown as mean + s.d. P adjusted values obtained after multiple testing correction ($P$adj < 0.01 was used as threshold of significance) are shown in **e**, **f**

a control reporter lacking the p53 3′UTR (Supplementary Fig. 5G). Mutation of the proximal and distal Tia1-binding sites in the p53 3′UTR increased translation of the reporter gene further (Fig. 6e) indicating that these sequences enabled translational regulation of p53 expression upon DNA damage induction.

**Etoposide induces IRES-mediated p53 translation**. Changes in transcription, translation and protein stabilization have been all linked to the regulation of p53 expression but the relative con- tribution of each of these processes to p53 induction in mouse B cells undergoing DNA damage is unknown. Thus, we investigated further which of these processes contributed to the increased expression of p53 in B cells following treatment with etoposide.

The inhibition of gene transcription with actinomycin D increased p53 protein expression to a similar extent as that induced by etoposide (Fig. 6d and Supplementary Fig. 6E). Actinomycin D induces DNA damage[35], but it did not affect SG assembly in activated B cells (Supplementary Figs. 1F, 6B). Thus induction of p53 protein expression in B cells upon etoposide treatment was independent of the novo p53 mRNA transcription. Next, we investigated whether proteasome-mediated degradation controls p53 expression levels in B cells. Inhibition of

proteasome-mediated protein degradation with MG132 or lactacystin did not increase overall p53 protein abundance in activated B cells as did etoposide or actinomycin D (Fig. 7a, b). Interaction of p53 with the E3 ligase mouse double minute 2 homolog (Mdm2) induces p53 protein ubiquitination and degradation. This interaction is tightly regulated, and destabiliza- tion of Mdm2 by the ubiquitin-specific-processing protease 7 (USP7) induces p53 expression and cell cycle arrest[36]. Inhibition of the Mdm2-p53 interaction with Nutlin-3 or the p53 Activator III RITA did not increase p53 protein levels in LPS-activated B cells (Fig. 7b, c). Etoposide inactivates Mdm2- mediated regulation of p53[37], therefore Mdm2 inhibition did not enhance p53 expression or activation following DNA damage as expected. Similar results were obtained when inhibitors of Mdmx or Usp7 were used, indicating that the Mdm2-p53 axis of regulation has a minor role in controlling p53 expression in activated B cells.

De novo translation of pre-existing mRNA was required for p53 protein synthesis during the DNA damage response. B-cell treatment with the translation elongation inhibitors cycloheximide and puromycin blocked p53 protein synthesis following DNA damage (Fig. 8a, b and Supplementary Fig. 7A). By contrast, inhibition of CAP-dependent translation initiation by selective targeting of eIF4E/eIF4G interaction with the

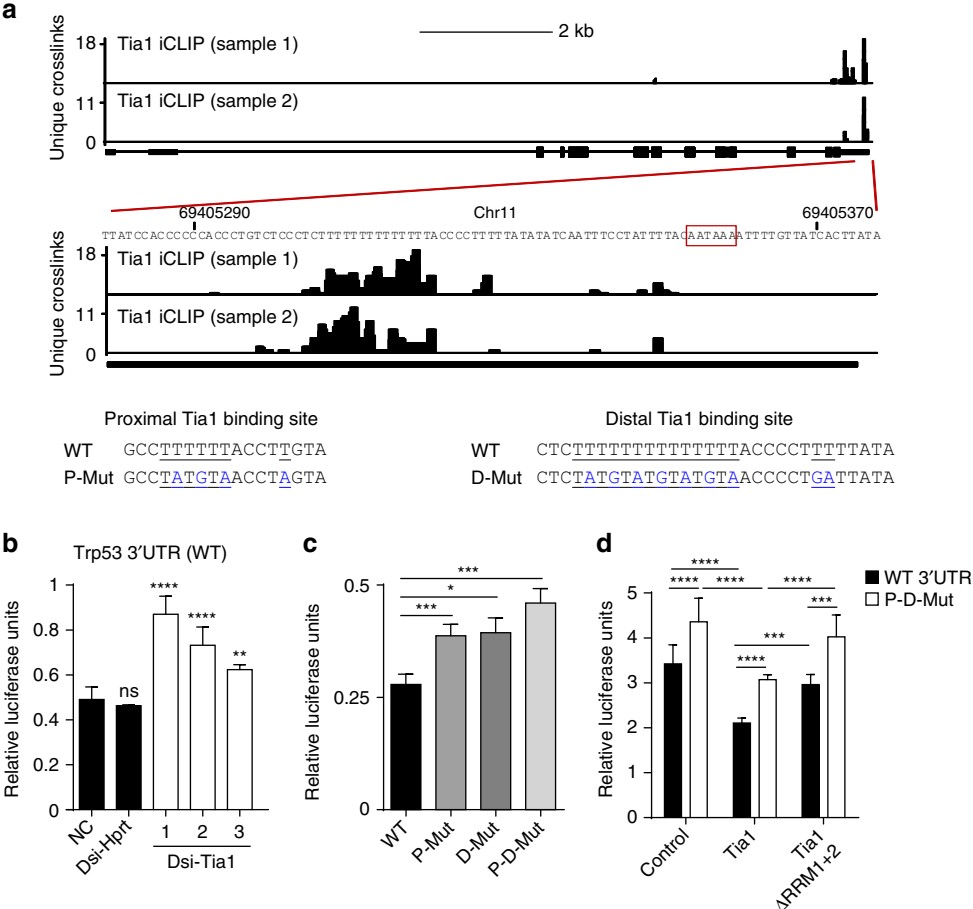

**Fig. 5** Tia1 binding to p53 mRNA silences p53 mRNA translation. **a** Map of Tia1-binding sites across the Trp53 gene locus. Tia1 binding to the 3′ terminal sequence of p53 3′UTR is magnified. *Red box* highlights p53 polyadenylation site. Data from two iCLIP experiments. *Bottom panels* show the nucleotide sequence of the proximal (closest to p53 stop codon) and the distal (closest to the polyadenylation site) U-rich Tia1 binding sites. Nucleotides in blue were substituted by site-directed mutagenesis for functional characterization. **b** Translation analysis of a renilla luciferase reporter construct containing the p53 3′UTR (psiCkeck2- p53 3′UTR) after knock down of Tia1 with three different DsiRNAs. A scramble DsiRNA was used as negative control. A Dsi-Hprt was used as positive control for cell transfection and gene knock down. Data from one of the three independent experiments performed are shown as relative luciferase units (RLU) (Mean + s.d., unpaired *t*-test, \*\**P* < 0.01, \*\*\**P* < 0.001, \*\*\*\**P* < 0.0001). **c** Functional analysis of the distal (D) and proximal (P) Tia1 binding sites in p53 3′UTR in mRNA translational regulation. RLU were quantified after 24-h transfection with a psiCkeck2- p53 3′UTR (WT), psiCkeck2- p53 3′UTR P-mutant (P-Mut), psiCkeck2- p53 3′UTR D-mutant (D-Mut) or psiCkeck2- p53 3′UTR P-D-mutant (P-D-Mut). Data from one of the five independent experiments are shown (Mean + s.d.; unpaired *t*-test, \**P* < 0.05, \*\*\**P* < 0.001). **d** Analysis of Tia1-dependent mRNA translational silencing. Cells were co-transfected with a psiCheck2-p53 3′UTR (WT) or a psiCkeck2- p53 3′UTR P-D-mutant (P-D-Mut) plasmid in combination with a MIGR1-Tia1, MIGR1-Tia1 ΔRRM1 + 2 or MIGR1-empty control plasmid. Data from the two independent experiments performed are shown as mean RLU + s.d. (RLU = renilla luciferase units divided by firefly luciferase units and normalized by the value from cells co-transfected with an empty vector) (Unpaired *t*-test, \*\*\**P* < 0.001, \*\*\*\**P* < 0.0001)

inhibitor 4EGI-1[38] did not bock p53 protein synthesis upon treatment with etoposide or actinomycin D (Fig. 8a, b and Supplementary Fig. 6A). Expression of Ccnd2, a well-known target of eIF4E, was reduced by 3-fold indicating effective inhibition of CAP-dependent translation by 4EGI-1 (Fig. 8b). Analysis of ribosome footprints mapped to the p53 mRNA indicated that ribosomes bind to an internal ribosome entry site (IRES) present in the 5′UTR of p53 in B cells (Fig. 8c). Notably, analysis of ribosome footprinting libraries generated from murine bone marrow dendritic cells, mouse embryonic fibroblasts and mouse liver cells showed the p53 IRES to be widely used in different cell types (Supplementary Fig. 5H). Analysis of ribosome footprints allowed us to identify the IRES nucleotide sequence (CTG) occupying the ribosome P-site (Fig. 8d and Supplementary Fig. 5H). To confirm that the p53 IRES was able to recruit ribosomes in a CAP-independent manner, we generated a

bicistronic luciferase reporter construct in which translation of a second open reading frame (ORF2), encoding the firefly luciferase protein, was controlled by the p53 5′UTR. Translation of ORF1 encoding renilla luciferase was measured as control. Transfection of HEK293 cells with this bicistronic reporter and measurement of luciferase activity showed that the p53 IRES sequence allowed translation of ORF2 (Fig. 8e). Mutation of the p53 IRES sequence reduced translation of ORF2 by 5-fold (Fig. 8e) without affecting mRNA abundance (Supplementary Fig. 5I). DNA damage induction with etoposide increased p53 IRES-dependent translation by 2-fold (Fig. 8f). Mutation of the p53 IRES prevented this increase in translation caused by etoposide without altering mRNA abundance (Fig 8f and Supplementary Fig. 5J), suggesting that the mouse p53 IRES is necessary for CAP- independent translation, as previously reported for human p53[18, 39]. Finally, we measured the renilla/firefly luciferase ORF ratio by qPCR as in some cases cloning of a

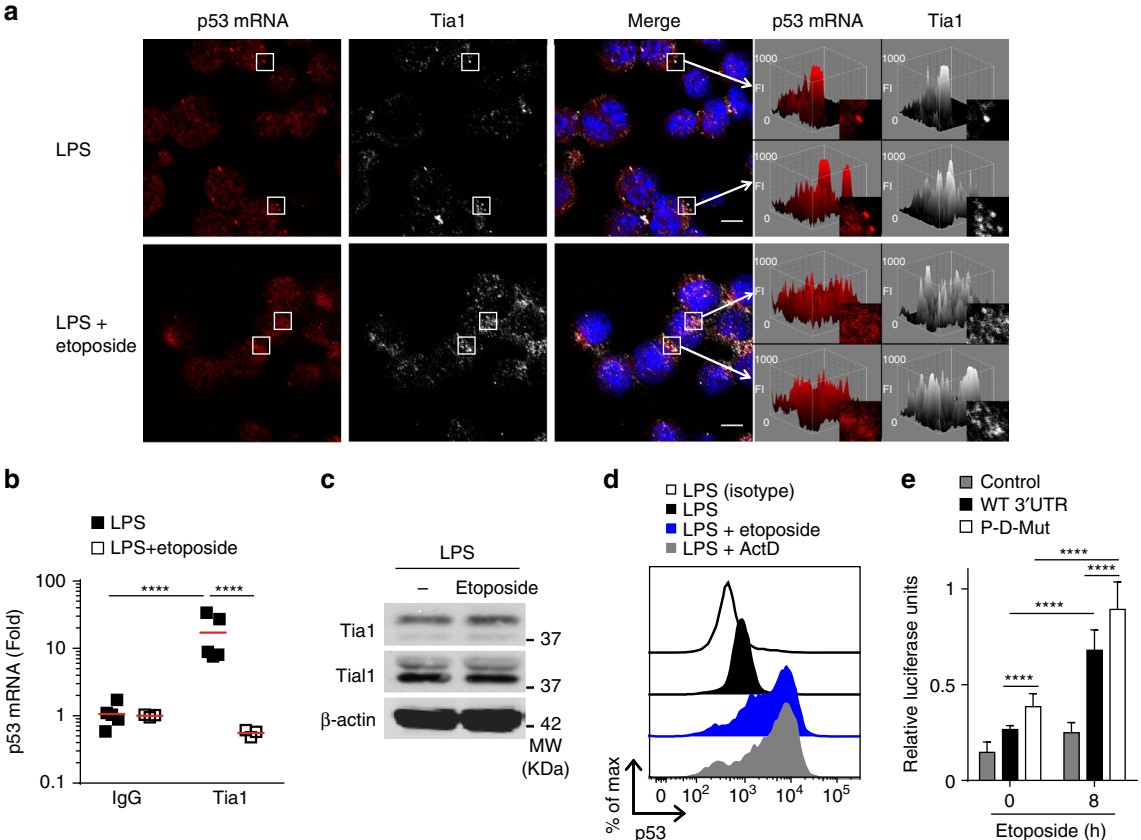

**Fig. 6** p53 mRNA is stored within stress granules. **a** Subcellular localization of p53 mRNA and Tia1 protein in LPS-activated B cells treated or not with etoposide. Fluorescence intensity 3D maps were generated for two regions (*white boxes*) to visualize Tia1 protein and p53 mRNA accumulation in cytoplasmic stress granules. *Z* axis of 3D maps relates to fluorescence intensity (FI). Representative images shown are from one of the two independent experiments performed in which a minimum of five confocal images were analyzed. Data extended in Supplementary Fig. 5a (*Scale bar* = 5 μm). **b** Tia1 protein: p53 mRNA co-immunoprecipitation in LPS-activated B cells treated or not with etoposide. An IgG isotype antibody was used as negative control in these RIP assays. Data are from two independent experiments. In each experiment, 1–3 experimental replicates were performed per condition (Mann–Whitney test, ****$P < 0.0001$). **c** Immunoblot analysis of Tia1, Tial1 and β-actin in B cells activated with LPS and treated or not with etoposide. **d** Analysis by intracellular flow cytometry of p53 protein expression in B cells activated with LPS for 48-h prior treatment with etoposide (20 μM) or actinomycin D (5 μg ml$^{-1}$) for 4 h. Data from one of the three independent experiments performed are shown in **c**, **d**. **e** Quantitation of renilla luciferase mRNA translation in HEK293 cells transfected with a psiCheck2 empty (control), a psiCheck2-p53 3′UTR (WT) or a psiCkeck2- p53 3′UTR P-D-mutant (P-D-Mut) vector and treated with etoposide (20 μM) for 8 h. Data from three independent experiments are shown as relative luciferase units (mean + s.d.; unpaired *t* test, ****$P < 0.0001$)

5′UTR between open reading frames may produce alternative transcripts due to differential splicing. The renilla/firefly luciferase ORF ratio was close to 1 in all conditions tested indicating that our bicistronic reporter constructs were not affected by alternative splicing (Supplementary Fig. 5I, J). Taken together, our data suggests that the regulation of translation, rather than transcription or protein stability, is responsible for p53 protein expression in B cells during the DNA damage response.

**ATM controls Tia1-mRNA dissociation and p53 translation**. Our data indicate that following DNA damage Tia1 is dissociated from p53 mRNA allowing the exit of p53 mRNA from SGs and its subsequent translation. This primary regulatory mechanism activates a p53-mediated genetic program that promotes cell cycle arrest, DNA damage repair and apoptosis[40]. The protein kinase ATM and its downstream targets Chk1 and Chk2 are essential for signalling control of the DNA damage response[41]. To gain mechanistic insight into how Tia1-mediated control of p53 expression is regulated, we selectively inhibited the enzymatic activity of these enzymes and measured p53 protein induction following DNA damage.

Inhibition of ATM (with KU55933) or both Chk1 and Chk2 (with AZD7762) prevented p53 induction after B cell treatment with etoposide (Fig. 9a, b and Supplementary Fig. 7B). Inhibition of Chk2 alone (with Chk2 inhibitor II) or inhibition of p38 MAPK (with SB203580), a kinase involved in the control of mRNA stability and translation, resulted in a minor reduction in p53 protein expression (Fig. 9a and Supplementary Fig. 7B). Inhibition of ATM or Chk1/2 kinases did not affect p53 mRNA abundance in B cells treated with etoposide compared to control cells (Fig. 9c and Supplementary Fig. 7C), but strongly affected p53 expression and activation as reflected by the reduced phosphorylation of p53 at Serine15 and by the reduced expression of Gadd45a and Cdkn1a (Fig. 9b). Etoposide-induced dissociation of Tia1:p53 mRNA complexes that triggers p53 mRNA translocation out of SGs and translation was reversed upon ATM kinase inhibition (Fig. 9d and Supplementary Fig. 7G). This was not due to differential Tia1 and Tial1 protein abundance after ATM kinase inhibition and DNA damage induction as these were unchanged in all conditions tested (Fig. 9b). This result led us to interrogate further the importance of ATM kinase during the DNA damage response in activated B cells. Global transcriptome analysis of B cells treated with ATM kinase

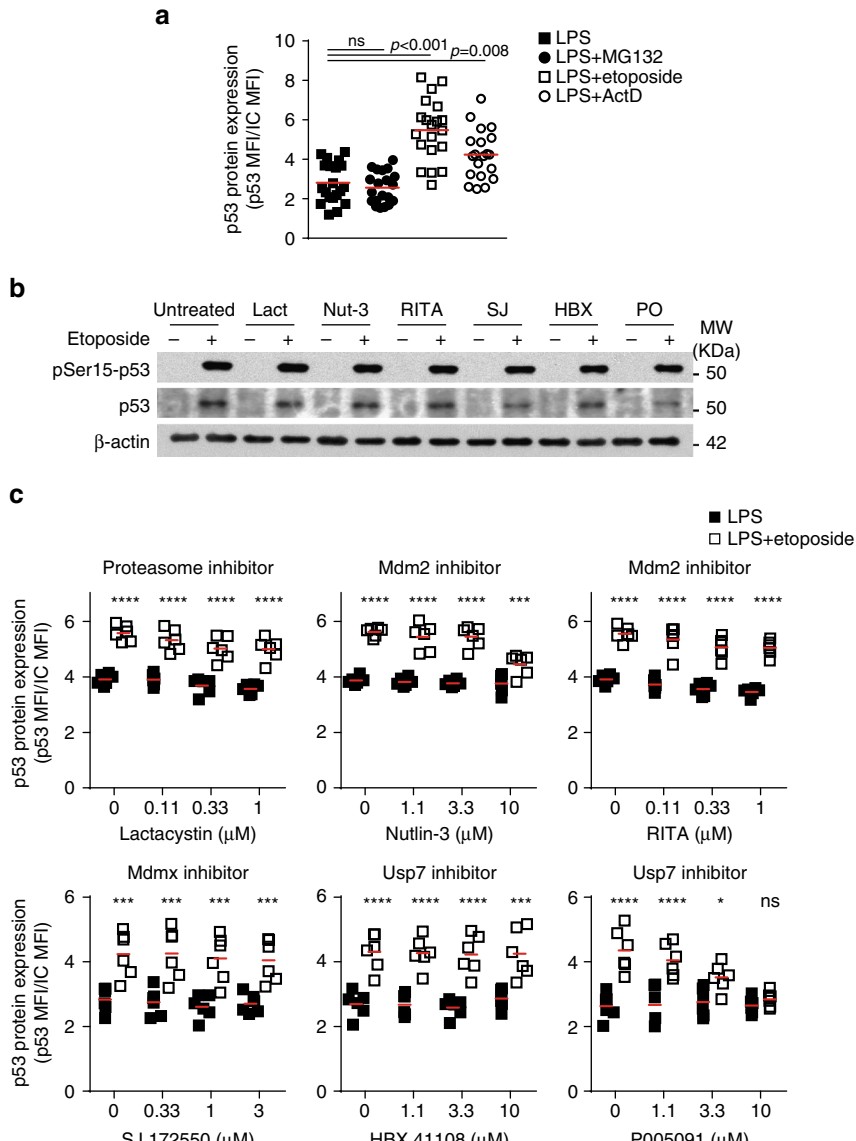

**Fig. 7** Proteasome does not regulate p53 protein synthesis in B cells. **a** Quantitation by flow cytometry of p53 protein expression in LPS-activated B cells treated with MG132 (10 μM), etoposide (20 μM) or actinomycin D (ActD, 5 μg ml$^{-1}$). Data are from six independent experiments with 2 to 4 biological replicates in each. Mann–Whitney test was performed. **b** Immunoblot analysis of pSer15-p53, p53 and β-actin in activated B cells treated with Lactacystin (Lact, 1 μM), Nutlin-3 (Nut-3, 10 μM), p53 Activator III RITA (1 μM), SJ172550 (SJ, 1 μM), HBX41108 (HBX, 10 μM) or P005091 (PO, 10 μM) prior induction of DNA damage with etoposide. One of the two independent experiments performed is shown. **c** Quantization by flow cytometry of p53 expression in B cells treated with different doses of the inhibitors described in **b**. Data are from six biological replicates collected in two independent experiments. Two-way ANOVA and Bonferroni post-test analysis was performed (*ns* non-significant, *$P < 0.05$, ***$P < 0.001$, ****$P < 0.0001$)

inhibitor KU55933 indicated a partial inhibition of etoposide-induced gene expression changes, although the transcriptome of these B cells was more similar to B cells treated with etoposide than to untreated control cells (Supplementary Fig. 7D, F). This close similarity was more pronounced when ribosome-associated mRNA was analyzed (Supplementary Fig. 6E), highlighting that a single cell-signalling module is not responsible for the complexity of the DNA damage response.

The ATM kinase-dependent interaction between Tia1 and p53 mRNA led us to investigate whether reduced p53 protein expression after cell treatment with KU55933 was due to restoration of Tia1-mediated mRNA translational silencing. Analysis of p53 mRNA distribution on polysomes showed that the accumulation of p53 mRNA in heavy polysomes of B cells treated with etoposide was significantly reduced upon inhibition of ATM kinase (Fig. 9e). Next, we asked whether this could be an

extended cellular mechanism that modulates protein synthesis in response to DNA damage and interrogated the polysome distribution of Hif1α and H2-K1, two other targets of Tia1 (Supplementary Fig. 8A, B) that were identified as having an increased translation once maintaining their mRNA abundance (Fig. 4d). Similarly, to p53, Hif1α and H2-K1 mRNA distribution in heavy polysomes significantly increased after B cell treatment with etoposide (Supplementary Fig. 8C, D). The polysome profile of Hif1α was restored upon ATM inhibition and, like p53, resembled the one in control B cells (Supplementary Fig. 8C). The effect of ATM inhibition in H2-K1 mRNA polysome distribution was inconclusive although it might suggest that H2-K1 mRNA shifts to fractions with a lower ribosome occupancy and less translation (Supplementary Fig. 8D). Taken together, our data shows that ATM kinase activation during the DNA damage response modulates Tia1 dissociation from its mRNA targets and

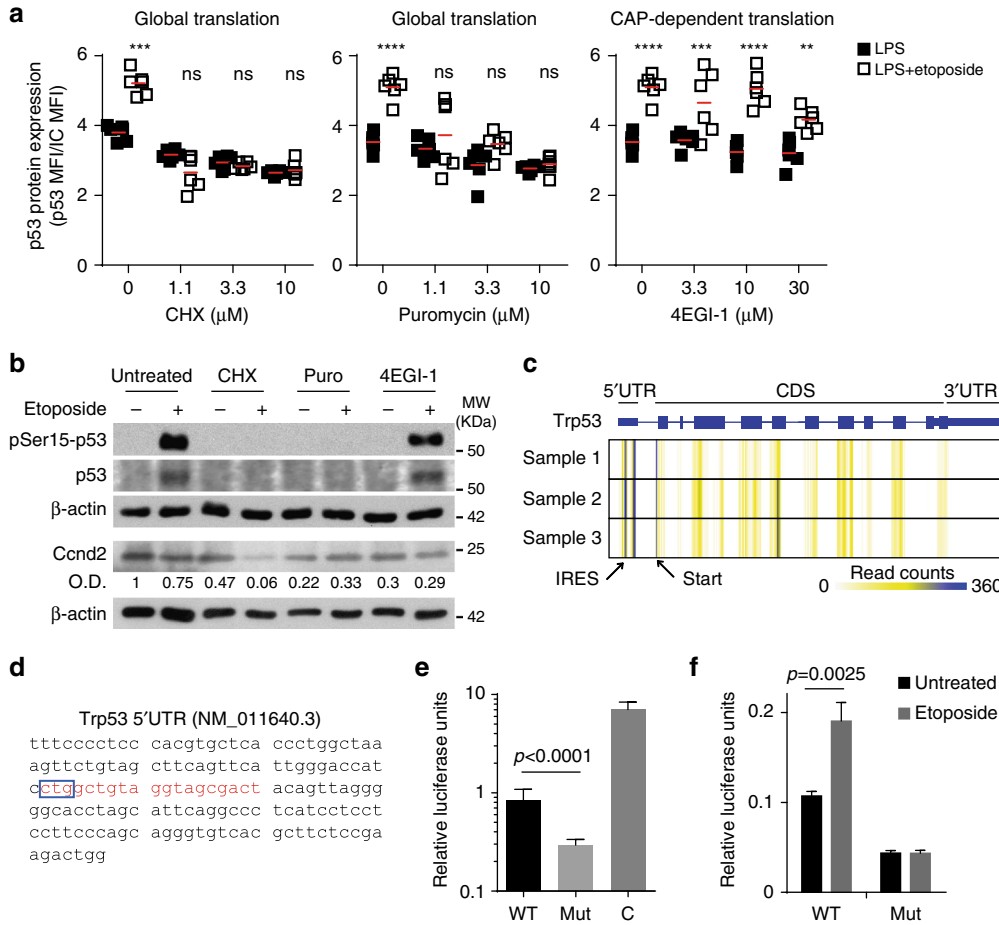

**Fig. 8** CAP-independent translation controls p53 protein synthesis in B cells upon DNA damage. **a** Flow cytometry analysis of p53 expression after inhibition of global translation with cycloheximide (CHX, 10 μM) or puromycin (Puro, 3 μM), or after inhibition of CAP-dependent translation with 4EGI-1 (10 μM). Data from six biological replicates collected in two independent experiments are shown. Two-way ANOVA and Bonferroni post-test analysis was performed (*ns* non-significant, **$P < 0.01$, ***$P < 0.001$, ****$P < 0.0001$). **b** Immunoblot analysis of pSer15-p53, p53, Ccnd2 and β-actin in B cells after inhibition of mRNA translation with CHX, puromycin or 4EGI-1. One of the two independent experiments performed is shown. Densitometry of Ccnd2 immunoblot after normalization by β-actin expression is shown relative to untreated cells. **c** Ribosome footprints mapped to p53 mRNA in LPS-activated B cells (GSE62148)[70]. Data from three biological replicates are shown as heatmaps. *Blue lines* mark ribosome footprint accumulation associated to the p53 ribosome entry site (reads in the first exon) and the first codon (reads spanning between the first and the second exon). **d** Murine p53 5′UTR nucleotide sequence cloned into the psiCheck2 plasmid to generate the bicistronic Trp53 IRES reporter construct. Sequence starting at the P-site (based on ribosome footprints; *blue box*) was mutated for Trp53 IRES characterization and it is shown in *red*. **e** Analysis of mRNA translation of the wild type (WT) or the IRES-mutated (mut) bicistronic Trp53 IRES reporter constructs. HEK293 cells were transfected in parallel with a psiCheck2 empty vector to produce monocistronic renilla luciferase and firefly luciferase mRNAs as a control **c**. Data from three independent experiments are shown as relative luciferase units (firefly luciferase units divided by the number of renilla luciferase units; mean + s.d.; unpaired *t* test). **f** Quantitation of p53 IRES-mediated translation upon induction of DNA damage. HEK293 cells were transfected with the Trp53 IRES reporter constructs described in **e**. After 24h, cells were treated with etoposide (20 μM) for 8 h. Data from three independent experiments are shown as relative luciferase units (firefly luciferase units divided by renilla luciferase units and normalized to the relative expression from a psiCheck2 empty vector; mean + s.d.; unpaired *t* test)

controls a translational program to produce key proteins that induce cell cycle arrest and promote DNA damage repair and apoptosis.

## Discussion

Programmed DNA damage is essential for lymphocyte development and antigen-mediated terminal differentiation into antibody-producing cells. Previous studies have focussed their attention on the characterization of pathways involved in permissive DNA repair of the immunoglobulin loci while preventing chromosomal translocation leading to tumour cell transformation[3, 4]. Here we show that post-transcriptional control of mature mRNA plays an important role in timely mRNA translation of key modulators of the DNA damage and stress response including p53 and Hif1α.

DDR is an extensively regulated process linked to cell cycle progression[5, 7] and B-cell differentiation[6]. Some estimates indicate that B lymphocytes undergoing programmed DNA damage (CSR and/or SHM) can divide within 5–8 h[42, 43]. However, gene expression from DNA transcription to mRNA and translation to protein occurs stochastically, and for low copy mRNAs significant oscillations and time delays affect the cellular protein content substantially[44, 45]. Temporal control of mRNA subcellular location in RNA granules and translation has been previously suggested to be an efficient mechanism for timely protein expression of cell cycle modulators such as Ccnb1[46]. Coordinated control of RNA regulons is important during cell cycle progression and DNA damage[47]. Increased expression of the protein components and assembly of PBs and SGs upon B-cell activation indicates that post-transcriptional regulation of mRNA expression is highly

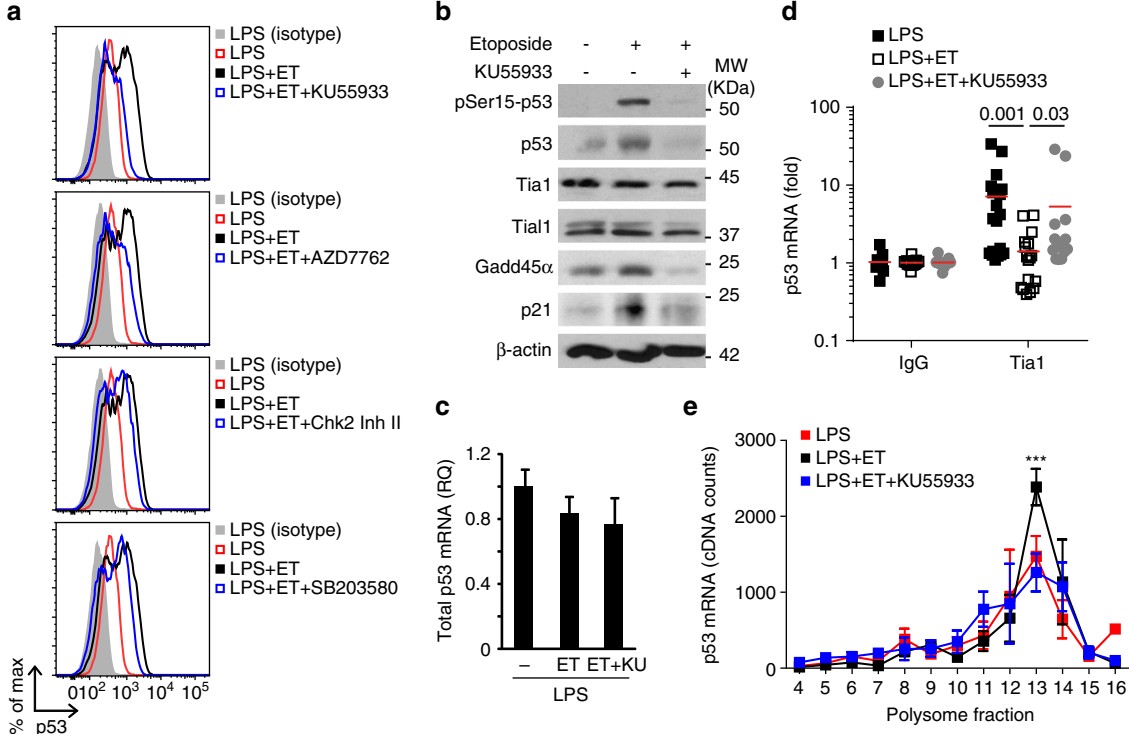

**Fig. 9** ATM kinase inhibition prevents Tia1 dissociation and p53 mRNA translation after DNA damage induction. **a** Flow cytometry analysis of p53 protein expression in B cells activated with LPS for 48 h and treated with etoposide (20 μM) for the last 4 h. ATM inhibitor KU55933, Chk1/2 inhibitor AZD7762, Chk2 inhibitor II and p38 inhibitor SB203580 (all at 10 μM) were added to the cells 1 h before treatment with etoposide. **b** Immunoblot analysis of pSer15-p53, p53, Tia1, Tia1, Gadd45a, p21 and β-actin in LPS-activated B cells incubated with KU55933 prior treatment with etoposide. One of the two independent experiments performed is shown. **c** Total cellular P53 mRNA abundance measured by RT-qPCR. Data are shown as mean + s.e.m. **d** Tia1 protein: p53 mRNA co-immunoprecipitation in LPS-activated B cells treated with etoposide in the presence of the ATM inhibitor KU55933. An IgG isotype antibody was used as negative control in these RIP assays. Data are from six independent experiments. 2–3 technical replicates were performed in each experiment. Mann–Whitney test was performed for statistical analysis and P values are shown. **e** Polysome profile of p53 from PolyRibo-3′ RNA-seq experiments performed in LPS-activated B cells incubated or not with the KU55933 inhibitor prior treatment with etoposide. Data are shown as mean + s.e.m. (n = 4, post hoc pairwise t-test, ***P < 0.001 for both comparisons: LPS vs. LPS + etoposide (LPS + ET) and LPS + Etoposide vs. LPS + Etoposide + KU55933 (LPS + ET + KU55933))

active in B lymphocytes. We have recently reported that post-transcriptional mRNA regulation by the RBP Zfp36l1 and Zfp36l2 is required for cell cycle arrest and effective VDJ recombination[48]. Zfp36l1 and Zfp36l2 proteins found within PBs promote active degradation of their mRNA targets[31]. Here we show that binding of the RBP Tia1 to specific targets, including p53 mRNA, regulates the residence of these mRNAs within SGs and their translational silencing. Our study supports the notion that p53 protein is not expressed in highly proliferative B cells[19]. Our data indicate that mRNA translational silencing may be rate limiting for p53 protein expression in this context.

The global characterization of Tia1:RNA interactions in B-cell extracts depleted of nuclei has revealed over 3000 genes whose transcripts have the potential of being translationally silenced within SGs. Further analysis of the RNA:protein interactions of other key components of SGs will increase our understanding of how these transcripts are regulated within these highly dynamic structures. Tia1 binds within 3′UTRs to U-rich elements similar to the binding motifs described in introns previously[49]. DNA damage induces Tia1 dissociation from its mRNA targets[50] and allows transcript release from SGs, polysome association and translation of selected regulators of the DDR. iCLIP data integration with global analysis of mRNA abundance and translation indicates that Tia1 mRNA targets involved in cell proliferation and maintenance are decreased upon DNA damage in B cells, consistent with previously described reduction in CAP-dependent translation[39, 51].

Translation of over 20% of Tia1 mRNA targets is increased after etoposide treatment. Comparison of changes in mRNA abundance and polysome association have allowed us to differentiate between genuine changes in translation from those caused by changes in mRNA abundance. Increased translation of p53, Hif1α and H2-K1 transcripts is mediated by CAP-independent translational mechanisms and are independent of "de novo" transcription. Internal ribosome entry sites (IRES) are present within p53 and Hif1α transcripts and are used as alternative translation initiation sites during stress responses[39, 52–54]. Inhibition of CAP-dependent translation does not prevent p53 protein synthesis in B cells following DNA damage and our data show that ribosome recognition of the p53 IRES is functional. Less is known about the translation of H2-K1 transcripts. However, a high degree of complexity is expected as different reports suggest that leucine-tRNA initiation at a CUG- non canonical start codon encodes for alternative peptides[55, 56].

For those Tia1 targets that showed increased mRNA abundance and translation during the genotoxic response we identified several genes that are regulated to some extent by p53 and Hif1α, including Cdkn1a and Gadd45a. These genes can also be post-transcriptionally regulated. Our iCLIP data reveal that both Cdkn1a and Gadd45a mRNA are targeted by Tia1 that likely controls mRNA translation of these regulators of cell growth and apoptosis during B cell activation. Recent evidence indicates that Cdkn1a mRNA can be recruited into arsenite-induced stress granules where its translation is silenced[57]. Similarly, Tia1

binding to Gadd45a mRNA silences translation[58, 59]. Post-transcriptional derepression of p53, Cdkn1a and Gadd45a mRNA by genotoxic stress allows synthesis of these proteins, with consequent cell cycle arrest and DDR. Tia1 and Tial1 dissociation from p53 mRNA in primary B cells is mediated by the ATM kinase and is less dependent on the MK2/p38 kinase axis that operates in p53-defective tumor cells[59].

ATM-dependent phosphorylation plays a major role in DDR. Phosphorylation of chromatin associated proteins including H2AX and 53BP1 marks double strand breaks in the genome, whereas phosphorylation of p53 mediates transcriptional activation and cell cycle arrest for DNA repair[60–62]. ATM activation controls AID-induced genotoxic stress required for B-cell differentiation in germinal centers[6] and ATM-deficiency results in deficient VDJ-recombination[63, 64], reduced CSR[10] and increased chromosomal translocation leading to lymphomagenesis[65, 66]. Our data provide evidence for another layer of control of the DDR controlled by ATM after B-cell activation. ATM-dependent dissociation of Tia1 and p53 mRNA release from SG after genotoxic stress is likely to be linked to p53 IRES folding, ribosome assembly and p53 mRNA translation[67]. We have not studied in detail the role of other protein kinases like ATR and DNA-PKs, but we anticipate that they can also control ribosome assembly and mRNA translation. We have gained insight here using a model that induces extensive DNA damage in primary cells and believe that this will prompt further studies to understand the importance of mRNA subcellular location and polysome assembly during physiological and pathological processes. Remarkably, our data suggest that post-transcriptional regulation rather than the proteasome controls p53 protein expression in activated B cells. This is an attractive and little explored mechanism that we are now starting to understand.

## Methods

**Antibodies and oligonucleotides**. Information is detailed in Supplementary Table 1.

**Animal Procedures**. C57BL/6 mice of 8–12 weeks old were used in this study. All animal procedures were approved by the Babraham Institute AWEEC and the UK Home Office.

**B cell isolation and cell culture**. Isolation of B cells from spleen or peripheral lymph nodes was performed by using the B Cell Isolation Kit from Miltenyi Biotec. B cells were cultured in RPMI 1640 Medium (Dutch Modification) plus 5% FCS, antibiotics, 2 mM L-glutamine and β-mercaptoethanol (5 μM). Etoposide (Sigma Aldrich, 20 μM) was added to the B cell cultures after 48 h stimulation with either LPS (Sigma Aldrich, E.coli 0127:B8, 10 μg ml[−1]) or with an anti-CD40 antibody (3/23 clone, 10 μg ml[−1]) plus recombinant mouse IL-4 (Peprotech, 10 ng ml[−1]) and recombinant mouse IL-5 (Sigma Aldrich, 5 ng ml[−1]). KU55933 (ATM inhibitor, Tocris, 10 μM), SB203580 (p38 inhibitor, Cell Signalling, 10 μM), Chk2 inhibitor II (Chk2 inhibitor, Sigma Aldrich, 10 μM), and AZD7762 (Chk1/2 inhibitor, Axon MedChem, 10 μM) were added, when indicated, 1-h prior treatment with etoposide. Actinomycin D (ActD, 5 μg ml[−1]), cyclohexamide (CHX, 100 μg ml[−1]), MG132 (10 μM), Puromycin and SJ172550 were purchased from Sigma Aldrich. HBX41108 and P005091 were purchased from Tocris. Lactacystin, Pifithrin-α, Nutlin-3, p53 Activator III RITA and 4EGI-1 were puchased from Calbiochem.

**Flow cytometry**. Intracellular staining of p53 was performed following indications from the manufacturer (BD Biosciences). Cytofix/Cytoperm and Fixation/Permeabilization solutions (BD Biosciences) were used for cell fixation and permeabilization as indicated. Data was acquired using a BD LSR II cytometer and analyzed using FlowJo X (TreeStar).

**RNA FISH and confocal microscopy**. For confocal microscopy, B cells were attached to coverslips that were treated with Poly-L-Lysine (Sigma Aldrich) for at least 1 h. at 37 °C. Ex vivo and activated B cells were cultured and treated as previously described. Fifteen to 30 min prior cell recovery, B cells (2–5 × 10[5]) were transferred to 24-well dishes containing the Poly-L-Lysine- coated coverslips. Cell adhesion was confirmed under the microscope prior washing unattached cells off with PBS (at 37 °C). Ice-cold methanol was added onto coverslips for B-cell fixation and permeabilization at −20 °C for 5 min. After drying, cells were fixed again with

4% paraformaldehyde in PBS for 15 min at RT. After extensive washing with washing buffer (PBS plus 0.3% Triton X-100), cells were blocked for 15 min using 1% BSA in washing buffer prior incubation with specific antibodies against Tia1, Dcp1a and eIF3η (see Supplementary Table 1 for more details). After washing, secondary antibodies coupled to Alexa Fluor 488, Alexa Fluor 555 or Alexa Fluor 647 (Life Technologies) were added. After 1-h incubation, samples were incubated with Dapi for 10 min and washed with PBS (three times), with H₂O (once) and mounted on microscope slides using Prolong (Life Technologies).

RNA FISH procedure was adapted from Widgerde et al.[68] Briefly, a 991 nucleotide- long sequence of the P53 mRNA coding sequence was used as template for generation of digoxigenin-labeled cDNA probes. Template sequence was amplified using specific B-cell mRNA, specific p53 PCR primers (forward—TGCTCACCCTGGCTAAAGTT; reverse: TCTTCTGTACGGCGGTCTCT) and Pfu Ultra II Fusion HS polymerase (Agilent). Then, it was cloned into a pCR2.1 TOPO® TA cloning vector following manufacturer indications (Life Technologies). Cloning orientation of P53 template sequence (5′ to 3′) was from the T7 promoter sequence to the M13 reverse priming site. Five microgram of linearized vector after BamHI enzymatic digestion was used as template for in vitro transcription using MAXIscript kit (Ambion) and T7 RNA polymerase (Promega). After DNAse I treatment, RNA was extracted by phenol/chlorophorm (pH 4.5) extraction followed by ethanol precipitation using ammonium acetate. Reverse transcription of RNA probe to generate antisense cDNA probes was performed using a Superscript III retro-transcriptase (Life Technologies) and digoxigenin (DIG)-11-2′-deoxy-uridine-5′-triphosphate (alkali-stable from Roche) for probe labelling. Treatment with RNAse H (Life Technologies) was performed before p53 cDNA probes purification with a Qiaquick Nucleotide Removal Kit (Qiagen). For RNA FISH, B cells were fixed and permeabilized with methanol followed further fixation with 4% paraformaldehyde as described before. After extensive washing with washing solution (0.1% Tween 20 in PBS), coverslips were treated with 70% ethanol for 5 min at 22 °C prior air drying. Samples were re-hydrated and incubated in blocking solution (0.1% Tween 20, 5% BSA fraction V or 5% non-reactive donkey serum in PBS) for 15 min. RNase inhibitors or Ribosyl Vanadyl Complex (10 mM, New England Biolabs) were added to all washing and blocking solutions. Then, samples were incubated with an anti-Tia1 primary antibody (Supplementary Table 1) for 2 h at room temperature. After washing, samples were re-fixed with 4% paraformaldehyde in PBS for 10 min. Samples were washed and quenched for 20 min with 0.1 M Glycine in washing solution. Then, they were incubated twice with 50% formamide/ 2× SSC (5 min each time) prior incubation with a p53 cDNA probe mix (100 μg ml[−1] DIG-labelled oligonucleotide probe, 50% formamide/2× SSC, 6% PEG, 0.2 mg ml[−1] BSA fraction V, 0.1 mg ml[−1] E. Coli RNase-free tRNA) that was previously denaturalized at 65 °C. Samples were incubated with the P53 cDNA probe mix over night at 37 °C in a humidified chamber. Then, they were sequentially washed with 50% formamide/ 2× SSC (twice for 10 min at 37 °C), with 50% formamide/ 1× SSC (twice for 15 min at 37 °C), with 1× SSC (once for 15 min at 22 °C), with 0.5× SSC (once for 15 min at 22 °C), with washing buffer (twice for 5 min at 22 °C) and blocking buffer (once for 15 min at 22 °C). P53 cDNA probes were detected with a sheep anti-DIG antibody coupled to AlexaFluor 555 (Molecular probes). Tia1 antibody was detected using a donkey anti-goat secondary antibody coupled to AlexaFluor 647. After 1 h incubation, samples were washed and mounted in microscope slides as indicated above.

All images were captured with an Olympus FV1000 System and processed using FV10-ASW 2.0 software. In all experiments, B cells isolated from 2–3 mice were treated and stained in parallel in each of the, at least, two independent experiments performed. Images were captured using 60× objective lens and ×4 digital magnification in order to analyze several cells per microscopy field. Images from at least three different microscopy fields were analyzed per replicate. The number of Tia1 and Dcp1a objects in each microscopy field was calculated based on florescence intensity signal using Volocity software and was divided by the number of cells in each field. Data are shown as mean + s.d. (n > 6 for all conditions tested). Digital magnification and 3D plot analysis of the fluorescence signal was performed with Image J software to visualize Tia1- and Dcp1a- containing RNA granules. Pearson R correlation of fluorescent signal colocalization was calculated with ImageJ.

**Plasmid constructs**. The psiCkeck2- p53 3′UTR plasmid was generated by PCR amplification of the mouse Trp53 3′UTR (445 bp) using B cell mRNA as cDNA template, Pfu Ultra II Fusion HS polymerase (Agilent) and specific primers (Supplementary Table 1), followed by cloning into a psiCkeck2 plasmid vector (Promega). XhoI and NotI restriction enzyme sites were added for directional cloning downstream the Renilla luciferase ORF. Mutagenesis of the proximal (P) (closest to the Trp53 stop codon) and/or distal (D) Tia1 binding sites (closest to the Trp53 PolyA signal) was performed using Gibson assembly to generate the following plasmid: psiCkeck2- p53 3′UTR P-mutant (P-Mut), psiCkeck2- p53 3′UTR D-mutant (D-Mut) and psiCkeck2- p53 3′UTR P-D-mutant (P-D-Mut).

The bicistronic Trp53 IRES vector was generated by substituting the polyadenylation site of the renilla luciferase and the Herpes simplex virus (HSV) thymidine kinase promoter, present upstream the firefly luciferase CDS in the psiCheck2 vector, with the mouse Trp53 5′UTR (NM_011640.3). Trp53 ribosome entry site was substituted by a SalI restriction enzyme site to generate the Trp53 IRES mutant vector.

Mouse Tia1 and a mutant Tia1 lacking the RNA-recognition motifs 1 and 2 were cloned into the MIGR1 plasmid vector after PCR amplification using specific primers (Supplementary Table 1). Restriction enzyme sites for BglII and EcoRI were added and used for directional cloning into the MIGRI plasmid vector.

**Luciferase reporter assays.** HEK293T (293T ATCC CRL-3216) cells, cultured in DMEM supplemented with 10% FCS, 2 mM L-glutamine and 1% Penicillin/Streptomycin, were transfected with the indicated plasmid constructs using Lipofectamine 2000 (ThermoFisher). Renilla and firefly luciferase expression was measured after 24 h with a Dual-Luciferase Reporter Assay System (Promega). Each independent experiment was performed in duplicates or triplicates. Data are shown as relative luciferase units. In those experiments assessing the role of p53 3′UTR in mRNA translation, relative luciferase units are calculated by dividing the number of renilla luciferase units by the number of firefly luciferase units (to control for changes due to different transfection efficiencies). To account for extrinsic changes in the Dual-Luciferase® Reporter Assay System due to cell treatment with etoposide, Tia1 over-expression or Tia1 knockdown, relative luciferase units were further normalized by the relative changes in renilla luciferase translation compared to firefly luciferase translation in cells transfected with a control psiCkeck2 empty vector. Trp53 IRES dependent translation was quantified by measuring firefly luciferase translation relative to control renilla luciferase translation. At least three independent experiments were performed and pulled data is shown as mean + s.d. Statistical analysis was performed using the unpaired Student's $t$-test.

**Dicer-substrate short interfering RNAs and gene knock down.** TriFECTa dsiRNA duplex kits containing three different Dicer-substrate short interfering RNAs (DsiRNAs) against Tia1 or Hprt were purchased from Integrated DNA Technologies (IDT). Hprt dsiRNA and scramble dsiRNA provided with the kit were used as experimental negative controls. DsiRNAs were transfected into HEK293 cells by using Lipofectamine RNAiMAX transfection reagent (Thermo-Fisher). After 48 h, plasmid and dsiRNA were co-transfected with using Lipofectamine 2000 (ThermoFisher). Gene knock down was evaluated 24-h after using qPCR and flow cytometry. Renilla and firefly luciferase expression was measured with a Dual-Luciferase Reporter Assay System (Promega).

**Polysome fractionation.** For polysome fractionation, B cells were incubated with cyclohexamide (CHX, Sigma Aldrich, 100 μg ml$^{-1}$) for 3 min at 37 °C prior collection. After washing the cells with ice-cold PBS containing CHX (100 μg ml$^{-1}$), cell pellets were either stored at −80 °C or resuspended in polysome buffer (300 mM NaCl, 15 mM MgCl$_2$, 15 mM Tris-HCl; pH 7.5, 100 μg ml$^{-1}$ cycloheximide CHX, 100 U RnaseOUT from Life Technologies) containing 1% Triton X-100. After incubation on ice for 10–15 s, cells were spun at $17,000 \times g$ at 4 °C. Cytoplasmic cell supernatant was placed at the top of a 50–10% sucrose gradient prepared previously in 17 ml- Beckman tubes (by underlying, from the bottom to the top, 3 ml of 10, 20, 30, 40 and 50% sucrose solutions made up in polysome buffer). Then, samples were centrifuged at $250,000 \times g$ for 2hr at 4 °C (acceleration 7, deceleration 7) using a pre-cooled SW41Ti. Samples were kept at 4 °C at all times. One millilitre fractions were automatically collected using gradient fractionator after injecting a 65% sucrose solution (made up in polysome buffer containing Bromophenol blue as a tracer dye) to the bottom of the tube. Polysome fractions were mixed with 3 ml of 7.7 M guanidine HCl and 4 ml of 100% ethanol, and incubated overnight at −20 °C.

**RNA extraction and RT-qPCR.** Ethanol precipitation procedure was follow for mRNA precipitation. mRNA was used for either 3′ RNA-seq library preparation (described below) or it was converted into cDNA to perform RT-qPCR analysis as follows[69]: Briefly, RNA retro-transcription (RT) was performed using SuperScript II (Life Technologies) and qPCR assays were performed using Platinum Quantitative PCR SuperMix-UDG or SYBR Green PCR Master Mix (Life Technologies). Total RNA extraction was performed using TriZol (Life Technologies) and converted into cDNA similarly. Primers and Taqman probes used are described in Supplementary Table 1. Quantification of cell mRNAs was calculated by the ΔΔCT (comparative threshold cycle) method, following manufacturer's instructions. 18S rRNA and β2m were tested to normalize mRNA abundance. Data are shown as relative quantification to the mRNA levels of either ex-vivo or LPS- stimulated B cells as indicated in each figure legend. Quantification of mRNA levels in each polysome fraction is shown as arbitrary units (a.u.).

**SDS-page and immunoblotting.** Protein from each polysome fraction was precipitated using trichloroacetic acid (TCA). Briefly, 1 volume of 20% TCA was added to each fraction. After incubation overnight at 4 °C, samples were spinned down at $17,000 \times g$. Pellet was washed twice with 0.5 ml of ice-cold acetone. After drying the pellet, protein was resuspended in sample reducing buffer (2×) containing 2-mercaptoethanol. 10% of the total was loaded in 10–12% polyacrylamide-SDS gels.

Total B cell protein extracts were prepared using RIPA buffer (50 mM Tris-HCl, pH 7.4, 150 mM NaCl, 1% NP-40, 0.1% SDS and 0.5% sodium deoxycholate) containing protease and phosphatase inhibitors (Protease inhibitor cocktail 3,

Sigma Aldrich). After 15 min at 4 °C, samples were centrifuged ($17,000 \times g$, 5 min) to isolate protein containing supernatants. Protein concentration was measured using a BCA protein assay (Pierce). 10% polyacrylamide-SDS gels were loaded with 20 μg. of protein lysate for electrophoresis. Proteins were then transferred to nitrocellulose membranes and immunoblotted with specific primary antibodies against p53, p53-Ser15, Cdkn1a p21, Gadd45a, Tia1, Tial1, eIF3η, G3BP1, PABP1, Dcp1a and β-actin (Antibody dilution 1:1000; see Supplementary Table 1). Specific HRP-conjugated secondary antibodies followed by enhanced chemiluminescence (Amersham Pharmacia Biotech) were used for detection. Uncropped immunoblot scans with molecular size markers are provided in the Supplementary Data.

**Individual-cross-linking and immunoprecipitation (iCLIP).** iCLIP experiments were performed with minor modifications as previously described[30, 70]. Briefly, primary splenic B cells were activated with LPS for 48 h prior UV- light crosslinking (245 nm light, 150 mJ cm$^{-2}$ using a Stratalinker 2400) of protein : RNA interactions. Cytoplasmic extracts were obtained after incubation with cytoplasmic extract lysis buffer (10 mM HEPES; pH 7.6, 10 mM KCl, 0.1 mM EDTA, 0.1 mM EGTA and protease inhibitors) as described above, and treated for 3 min at 37 °C with a high concentration (12 U ml$^{-1}$) or a low concentration of RNase I (0.167 U ml$^{-1}$). Tia1:RNA complexes were immunoprecipitated using 2 μg an anti-Tia1-specific antibody (Supplementary Table 1) coupled to protein G Dynabeads. An isotype IgG antibody was used as negative control and specific Tia1 immunoprecipitation was checked by western blot. After immunoprecipitation, beads were washed twice with high-salt buffer (50 mM Tris-HCl, pH 7.4, 1 M NaCl, 1 mM EDTA, 1% NP-40, 0.1% SDS and 0.5% sodium deoxycholate) and twice with PNK washing buffer (20 mM Tris-HCl, pH 7.4, 10 mM MgCl2 and 0.2% Tween-20). One tenth of each sample was labelled with $^{32}$P-ATP using PNK. The rest was ligated to an RNA linker after RNA 3′ dephosphorylation. RNA–protein complexes were separated by SDS-PAGE and were transferred to nitrocellulose membranes. Tia1-RNA complexes with a molecular size above 55 KD were isolated after visualization by autoradiography. RNA extraction from the nitrocellulose was performed with proteinase K in 200 μl of PK buffer (100 mM Tris-HCl, pH 7.5, 50 mM NaCl and 10 mM EDTA) for 10 min at 37 °C. After that, additional 200 μl of PK buffer containing urea (7 M) were added to each sample that was further incubated for 20 min at 37 °C. RNA was isolated by phenol-chloroform extraction and ethanol precipitation, and retrotranscribed into cDNA using RCLIP primers (Supplementary Table 1) and SuperScript III. cDNA purification was performed using 6% TBE-urea gels and was followed by cDNA circularization and amplification by PCR using Solexa P5/P3 primers. cDNA libraries were analyzed prior Illumina (GaIIX) sequencing. RCLIP primers include a seven-base-long 'barcode' at the 5′ end (three known bases plus four unknown nucleotides) for sample multiplexing and PCR duplicate read identification. Two known bases (AT, 3′ barcode) were added to the 3′end of each RNA molecule by the RCLIP primers.

**RNA immunoprecipitation (RIP).** Validation of Tia1 and Tial1 interaction with selected mRNA targets was performed by RIP as follows[69]: Briefly, RIP assays were performed with cytoplasmic cell extracts were obtained using cytoplasmic cell lysis buffer (10 mM TrisHCl; pH 7.4, 10 mM NaCl, 2.5 mM MgCl$_2$, 40 μg ml$^{-1}$ digitonin from Sigma Aldrich) containing protease inhibitors (p8340, Sigma Aldrich) and RNase inhibitors (40 U ml$^{-1}$ of RNase Out, Life Technologies). After incubation on ice for 15 min, 10 μl of 10% NP-40 was added. Samples were then vortex for 10 s and centrifuged at $900 \times g$ at 4 °C for 20 min. Supernatant was collected as the cytoplasmic fraction. Protein A/G magnetic beads (Pierce, ThermoFisher Scientific) were coupled with 2 μg of antibody against Tia1 or Tial1 or an isotype control IgG antibody (Supplementary Table 1). After 1-h incubation at room temperature, beads were washed with NET-2 buffer (50 mM Tris-HCl pH 7.5, 500 mM NaCl, 0.05% NP-40 and 1 mM MgCl$_2$). Two hundred fifty microgram of cytoplasmic protein extracts were added to the antibody-coupled beads for immunoprecipitation (overnight at 4 °C in rotation). Then, beads were washed (5×) with NET-2 buffer and proteins were digested with proteinase K (0.5 mg ml$^{-1}$, Roche) dissolved in NET-2 buffer containing 0.1% SDS. After 15 min at 55 °C, RNA was extracted by adding an equal volume of phenol: chloroform: isoamyl alcohol (25:24:1, Sigma, P3803). After mixing, centrifugation and isolation of the hydrophilic sample phase, RNA was precipitated using 0.1 volumes of 3 M NaOAc, 0.3 volumes of 1 mM EDTA and 2.5 volumes of 100% ethanol. RNA pellets were washed twice with 70% ethanol prior drying and re-suspension in water. Reverse transcription into cDNA and qPCR assays were performed as described above. Data from RNA immunoprecipitation assays are shown as fold enrichment relative to LPS- only treated B cell samples. In all experiments, samples from two or three biological replicates were processed in parallel to determine biological variability. Data are shown as Mean + s.d.

**Library preparation and high throughput sequencing.** mRNAseq libraries were generated with a TruSeq Stranded mRNA Sample Prep Kit (Illumina). mRNAseq and iCLIP cDNA libraries were sequenced with a HiSeq2000 Illumina system (mRNAseq—100 bases, single end; iCLIP—50 bases, single end). mRNAseq data from naïve and LPS-activated B cells were obtained from publically available data sets (GSE62129). cDNA libraries for PolyRibo-3′ RNA-sequencing was performed using identical concentrations from each of the RNA sample fractions collected after polysome fractionation that was performed as indicated above. Between 62.5

and 625 ng of RNA was retro-transcribed using one specific RmRNAt primer with a unique 10-bases barcode that allows library multiplexing (41 libraries were multiplexed in each sequencing lane) and PCR deduplication. Of the 10-bases barcodes, 5 nucleotides were known (for library multiplexing) and 5 unknown (for PCR deduplication) (Supplementary Table 1). After cDNA isolation and purification, input material was amplified using the MessageAmp II aRNA Amplification kit. Briefly, a second DNA strand was generated. Double strand DNA was purified using cDNA Filter Cartridges and in vitro transcription was then performed using T7 RNA polymerase. After DNA degradation using DNAse I, amplified in-vitro transcribed RNA was purified using the aRNA filter cartridges provided. Then a second retro-transcription was performed using the 2RT primer. After RNA digestion with RNAse A, cDNA of a size between 110–150 bases was purified using TBE-Urea gels and used as template for PCR library amplification using P5/P3 Solexa primers. cDNA libraries were then sequenced using a HiSeq1000 Illumina system. Paired sequencing was performed (10 cycles from the 5′ end to read unique barcode and 60 cycles from the 3′end). Ribosome footprinting libraries have been described previously (GSE62148)[70].

**Bioinformatics and statistics**. FastQC was used for quality analysis of sequencing data. Data analysis was done in the iCount pipeline[30]. Read mapping and peak call analysis of Tia1 iCLIP data[30, 70] was done as follows: Briefly, sample demultiplexing was performed based on the three known bases present at the 5′end of each read. The four unknown bases of the 5′ 'barcode' were used for removal of PCR duplicate reads. The length of unique reads was defined by a two bases (AT)- long 3′ barcode. Reads were then trimmed to remove adaptor and barcode sequences before mapping to the genome with Bowtie. Mouse genome assembly GRCm37 (NCBI; UCSC mm9 annotation) was used originally. Data were lifted over to GRCm38 (NCBI) assembly if required by using Galaxy. Nucleotide position −1 (relative to the first base annotated, set as position 0) of each read defines Tia1-crosslink–binding site. The number of unique cDNA counts associated to a Tia1-crosslink site is defined by both read length and the uniqueness of the 5′ 'barcode' sequence. Peak enrichment analysis was performed in iCount by genome-wide analysis of 30 base windows (15 nts. upstream and downstream from each mapped Tia1 crosslink site), allowing 100 permutations and multiple-hypothesis testing to correct for multiple comparisons. All Tia1- crosslink sites with a false-discovery rate (FDR) higher than 0.05 were not further considered. Tia1:RNA binding sites were associated to a particular genomic feature (5′UTR, 3′UTR, ORF, introns, ncRNA or intergenic—mm9 annotation-) and visualized in the UCSC genome viewer. Gene ontology (GO) analysis was performed using GOrilla and REViGO. GO terms related to biological processes (BP) were used.

mRNAseq data were mapped with TopHat2 and counted using HTSeq (Mus_musculus.GRCm38.78.gtf). Differential expression analysis was performed with DESeq2, that adjusted the data to a negative binomial distribution after correction by library size factor. Benjamini-Hochberg multiple test correction of the $P$ values was performed to calculate adjusted $P$ values ($Padj < 0.01$ was used as threshold of significance).

Data from PolyRibo-3′ RNA-sequencing was mapped to GRCm38/mm10 genome (gene and transcript annotation Ensembl 66.8) after demultiplexing and PCR deduplication based on 5′ end barcodes annotated in the 10 bp-first read of the Illumina run (see Supplementary Table 1 from RmRNAt primer sequence). Location and length analysis of the longest consecutive A tract in raw reads from the 60bp-second read of the Illumina run (deposited in GEO, see accession numbers below) was done first. Removal of the A-tract from reads was performed according to the following rules (tail length, A tract length, allowed mismatches in tract): (20, 17, 2), (19, 16, 2), (18, 15, 2), (17, 14, 2), (16, 13, 1), (15, 12, 1), (14, 11, 1), (13, 10, 1), (12, 10, 1), (11, 9, 1), (10, 8, 1), (9, 8, 1), (8, 7, 0), (7, 6, 0), (6, 5, 0), (5, 3, 0), (4, 3, 0). Reads containing internal A-tracts were cut according to the rule (minimal A tract length, allowed mismatches in tract): (17, 2). Finally, any remaining A at the end of the read was removed prior alignment to the reference genome and counting. Raw counts were used to generate a sample data matrix for DESeq2 to identified changes in the polysome fraction profile of each gene. First, a generalized linear model (GLM; read.counts ~ condition + polysome.fraction + condition:polysome.fraction) was estimated for polysome data and, second, the likelihood ratio test was performed between the original and a reduced (read.counts ~ polysome.fraction) GLMs. Then, we used a post-hoc pairwise t-test to find genes with the most significant fraction change between conditions in monosome, light and heavy fractions. The significance $P$-value threshold used in the pairwise t-test was 0.01. Although 3′RNA sequencing allows identification of alternative polyA site usage, it does not enable to identify and quantify alternatively spliced mRNA transcripts. Therefore, counts mapped to an Ensembl-defined gene unit were summed up and analyzed. The same approach was taken for quantifying mRNA abundance by mRNAseq.

mRNA and PolyRibo-3′ RNA-sequencing data from 4 biological replicates were collected to estimate variation and ensure reproducibility. Analysis of variance (ANOVA) followed by Bonferroni's correction, unpaired Student's t-tests or Mann-Whitney tests were performed for statistical analysis of non-sequencing data as indicated in each figure legend.

**Data availability**. The data sets generated during and/or analyzed during the current study are available in Gene Expression Omnibus. Accession codes are:

GSE62148 and GSE93576 (subseries GSE93573, GSE93574 and GSE93575). All other data are available upon request.

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

## Acknowledgements

We thank K. Bates, S. Walker, K. Tabbada, S. Andrews, R. Walker, S. Schoenfelder and the Babraham Institute Biological Support Unit for technical assistance. We also thank S. Cook, E. Monzon-Casanova, K. Okkenhaug for critical reading of the manuscript. This research was supported by Biotechnology and Biological Sciences Research Council (BBSRC) grants BB/J001457/1, BB/I003428/1 and BB/J00152X/1 and a Wellcome Trust Investigator Award to MT.

## Author contributions

M.D.D.-M. designed, planned and performed all the experiments with guidance from M.T.; M.D.D.-M. and J.U. designed and performed PolyRibo-3′ RNA-seq experiments. M.D.D.-M., V.Y.K, N.L.N., T.C. and J.U. performed the bioinformatics analysis. M.D.D.-M. analyzed and assembled all other data. M.T. obtained the funding. M.D.D.-M. and M.T. wrote the manuscript. All authors read and contributed in editing the manuscript.

## Additional information

**Competing interests:** The authors declare that they have no competing interests.

