## [Peer Review File · Nature Communications]

Reviewers' Comments:

Reviewer #1 (Remarks to the Author)

Diaz-Munoz and colleagues study Tia1, a translational silencer that can localize to stress granules (SGs), in post-transcriptional regulation of the DNA damage response during B cell activation. Studies of mRNP granules are important in post-transcriptionally regulating mRNAs but are not new; what is new here is the context during B cell activation and DNA damage, as can occur physiologically by DNA replication, CSR, and SHM in activated B cells. The authors argue that mRNA translation of pre-existing mRNAs is a key regulatory mechanism in B cell activation, which is at least partly controlled through the sequestration of Tia1-bound mRNA targets in SGs in an ATM-dependent release mechanism. They provide evidence that Tia1 binds to the 3'UTR of p53 mRNA to sequester it in SGs and suppress p53 mRNA translation. The release of p53 from SGs depended on activation of ATM kinase sensor of the DNA damage response. A list of Tia1 mRNA targets is identified in activated B cells, which may be part of an unappreciated program that regulates B cell development through translational repression and release with DNA damage, such as from CSR/SHM.

The study reports an interesting mechanism for the the rapid translation of DNA damage modulators in activated B cells, which provides context for SGs beyond model systems. The story is appealing but physiological relevance to implicate mRNA-Tia1 SG sequestration and release with translation as a significant mechanism regulating B cell development and differentiation appears incomplete at this stage and additional issues to be addressed arose during review including:

1. Page 4 text and Figure S1F- The formation of Tia1 containing SGs in activated B cells suggested that translational silencing occurs during B cell activation based on co-localization with eukaryotic initiation factor eIF3 η . However, not all Tia1-positive SGs co-localize with eIF3 η , and complex co-localization does not impute function. Has the association of SGs with polysomes for translation been examined or excluded in activated B cells?
2. Figures 3E and S3C, S3D- p53 mRNA did not increase with B cell activation and massive DNA damage with etoposide, although the heavy polysome associated p53 mRNA fraction increased. Gadd45a and Cdkn1a, known p53-dependent target genes, were induced. Was Gadd45a and Cdkn1a gene induction blocked with sh-p53 or other methods that impair p53 translation, stabilization, or nuclear localization? The data seems incomplete to impute increased p53 translation in activating these genes without showing increased p53.
3. Figure 3H- p53 co-localizes with Tia1-positive SGs in activated B cells and DNA damage with etoposide releases p53 from Tia1. However, confocal microscopy does not convincingly support this claim. Significant overlap between p53 mRNA and Tia1 protein appears to remain after etoposide treatment. Additionally, accumulation of Tia1 and p53 mRNA in SGs should be quantified, image resolution should be improved, and axis labels should be added on the fluorescence intensity 3D maps to clarify the data.
4. Figures 3I, S3G- are these replicates of the same experiment? Is the Figure S3G right panel a replicate experiment of Figure S3F?
5. Figure 3K, 3L- while statistically significant, the luciferase reporter repression assays from the p53 3'UTR with or without the Tia1 binding site are quite modest at <2-fold up to 3-fold changes. Combined with Figure S3C inability to show changes in p53 protein levels with activation and more physiologic levels of DNA damage from CSR/SHM, this does not give much confidence that this is a major regulatory mechanism for B cell development with activation and DNA damage response.

6. Figure S4A- are the p53 protein level changes statistically significant to make the claim that CHX blocked p53 translation from pre-existing mRNAs with etoposide or actinomycin-D?

7. Text lines 215, 216- the claim is that activated B cells translate p53 from pre-existing mRNA pools blocked by Tia1 in SGs once DNA damage has caused the dissociation to enable cell cycle arrest, DNA damage repair or apoptosis. However, the DNA damage induced in this study is well beyond physiologic levels with targeted CSR and SHM limited mainly to IG genes, and B cells activated by LPS or CD40/IL4/IL5, undergo CSR/SHM and cycle rapidly while dealing with limited IG locus repair. Therefore, it is not clear whether this mechanism is physiologically relevant during B cell activation. It is a key point to address whether loss of Tia1 function in activated B cells can induce premature p53 activation, cell cycle arrest and whether gain of Tia function can inhibit DNA damage repair and apoptosis. Additionally, the requirement of Tia1 in promoting efficient CSR and plasma cell differentiation is also key to address to determine the physiological relevance of Tia1 during later stages of B cell development focused on here.

8. In Figure 4D, the interaction between Tia1 and p53 mRNA is suppressed by ATM kinase activation during the DNA damage response. Based on this model inhibition of ATM kinase after induction of DNA damage should restore the physical interaction between Tia1 and p53 mRNA. There is no statistical significance for data presented in Figure 4D which raises the question whether the interaction between Tia1 and p53 mRNA is actually ATM kinase dependent.

9. The claim that Tia1 can globally control translation in response to DNA damage seems an overreach as the inhibition of ATM kinase did not suppress translation/polysome accumulation of all Tia1-mRNA targets after induction of DNA damage. There appeared to be no effect on H2-K1 translation after ATM inhibition in Figure S5D, which was identified as a top Tia1 mRNA target in Figure 3D.

Reviewer #2 (Remarks to the Author)

Regulation of mRNA translation and subcellular location controls protein synthesis of key modulators of the DNA damage response during B cell activation
Diaz-Munoz MD et al (Turner M corresponding)

Lymphocyte development into antibody producing cells involves programmed DNA damages for generation of diverse B cell receptors. This manuscript addresses the roles of stress granules and translational control in B lymphocyte activation and DNA damage response. Using CLIP, RNA-seq and Ribo-seq analyses, this manuscript suggests that in response to B cell activation with mitogens, there is formation of RNA stress granules that are critical for translation repression of a large collection of mRNAs with shared U-tracts in their 3'-UTR. Importantly, DNA damage triggers dissociation of p53 mRNA from Tia1 protein in the RNA granules, facilitating robust ribosome translation of p53 mRNA. Increased expression of p53 would then direct genetic programs that feature cell cycle arrest, DNA damage repair, and regulation of apoptosis. Release of p53 from stress granules involves ATM phosphorylation of p53 by mechanisms not yet resolved. Release of the p53 from stress granules was shown to involve the 3'-UTR portion of the p53 transcription. Robust translation of p53 mRNA released from stress granules during B cell development and DNA repair is suggested to involve an IRES embedded within the 5'-portion of the p53 mRNA, although this idea was not tested in the manuscript.

Overall, this manuscript addresses significant ideas concerning the roles of translational control in lymphocyte development. The central role of p53 trafficking to and from RNA granules during this process are novel, and the manuscript features large scale analyses of mRNAs localized to stress granules and their mRNA and translation changes during the DNA damage response. The Introduction of the manuscript is clearly written, and the experiments presented generally flow logically. The primary concern with the manuscript is that there are some conclusions that are not clearly drawn from experiments. Furthermore, there is uncertainty about some of the quantitation

presented in the Results. Finally, the manuscript would be bolstered by testing key features of the p53 model. Clarifying these experimental results and interpretation and testing key features of the p53 model are important to support the stated conclusions.

Reviewer concerns:

1. Figure 2, line 133: This section discusses a polysome/RNA-Seq analysis and highlights the importance of translational control in the activation of B cell undergoing a DNA damage response. There is a discussion of gene numbers and percent of those that are differentially expressed upon etoposide treatment that is difficult to follow. The reasoning and the justification for the conclusions are not clear to the reader. It is stated that there are 6340 gene differentially expressed. The term differentially expressed is very general and infers those with mRNA and/or translation changes involving either repression or activation. It is stated that "53% of these genes showed no significant changes in the amount of mRNA associated with ribosome or their distribution along polysomes (Fig. 2D, blot dots), suggesting that changes in mRNA abundance are not always linked to changes in translation." This sentence is not clear regarding both the reasoning and conclusion. This section continues with the statement that 3580 genes were differentially translated upon DNA damage induction (Fig. 2B and supplementary Table 3)." If 53% of the 6340 genes that were significantly measured showed no changes in the amount of mRNA associated with ribosomes, how would 3850 genes (~61%) be differentially translated? This section continues and states that "83% of these genes were differentially expressed, linking their cellular mRNA abundance with translation." This sentence is not clear regarding the meaning or quantitation. The final conclusion at line 153 is also problematic: "Taken together, these results show that the expression of a significant fraction of genes in activated B cells is independent of the overall cellular mRNA content and relies on post-transcription regulation of mRNA translation." It is not clear how one can conclude that the overall cellular mRNA content is not critical to the activation of B cells.

2. Figure 2, line 156 concerns the fate of the Tia1 associated transcripts. This section begins with the statement that "77% of Tia1 mRNA targets were differentially expressed, differentially translated or both following etoposide treatment. One typically views gene expression to involve transcription, mRNA decay, mRNA translation, and proteolysis, which collectively determine the level of protein expressed from a given gene. Furthermore, the emphasis here is on differential repression/activation of gene expression that occurs with mRNA trafficking to RNA granules, and one would argue this presentation should be more clearly delineate those with lowered translation or reduced mRNA levels from those being induced. Fig. 2G could be explained more clearly, and it is not clear how 20% of Tia1 mRNA targets were increased upon etoposide treatment from the data presented in Fig. 2H. In this case and many others, there is succinct conclusions drawn and quantitation without adequate reasoning that is discussed in the text. It is acknowledged that journal formats can limit word counts, but clarity of reasoning is imperative.

3. Figures 3 and 4: What are the numbers 1-4 representing at the bottom of Fig. 3D? Fig. 3M was not fully clear regarding the experiment arrangement. In Fig. 3F and other polysome profiling experiments that follow, the described normalization (18S rRNA and beta2m transcript) does not appear to be appropriate. For example, in Fig. 4E there appears to be an increase in polysome association of p53 mRNA upon etoposide treatment that is lowered by ATM inhibitor. However, the pattern of mRNA distribution does not show a shift in mRNA from polysomes to lighter fractions, rather just lowered transcript levels, which is inconsistent with experiments presented in the manuscript. This reinforces the importance of appropriate normalization (e.g. spike in controls or others) in the polysome profiling.

4. The underlying model argues that key sequences in the 3'-UTR of p53 transcript allows for association and controlled release of this transcript from the RNA granules during the DNA damage response. There is speculation that there is competition between Tia1 and one or more RNA binding proteins that associate with p53 and facilitate this release and ATM is involved in this process. How would ATM, and possibly p53 protein phosphorylation, fit into this model? How would

omission of the proposed Tia1 binding site in the p53 mRNA block trafficking to RNA granules and lead to elevated p53 mRNA translation even in the absence of the DNA damage response? What consequence would this have on the B cell processes? Some test of the key tenets of the model and its consequences on B cell fate would strengthen this manuscript.

Minor:

The abstract mentions protein translation, which should be referred to as mRNA translation or protein synthesis.

Reviewer #3 (Remarks to the Author)

The authors describe a study to identify mRNAs that interact with TIA1 in stress granules and may therefore be subject to regulation of translation. They follow up primarily on Trp53. They argue that p53 mRNA is highly expressed in developing B lymphocytes, but the mRNA is repressed by interaction with TIA1. This binding is released upon DNA damage induction, followed by translation.

The authors do a good job analyzing the stress granule associated mRNAs and changes in abundance of mRNAs in general and in the polysome fraction upon B lymphocyte differentiation and DNA damage induction. However, the part on regulation of mRNA expression at the level of translation is less clear. The model is very attractive: Trp53 mRNA is sequestered and only translated when needed. However, the data do not fully support this and are definitely not conclusive, as outlined below. It is an interesting idea, but will need much better experimental data before it can be considered a credible new mechanism of Trp53 regulation.

The widely accepted model for p53 protein regulation mainly considers degradation via hdm2 (or mdm2 in the mouse) when no DNA damage is present. Upon DNA damage induction, p53 is phosphorylated and loses its affinity for hdm2, resulting in protein stabilization, because it is no longer degraded by the proteasome. The authors do very little to investigate this mechanism. The only thing they try is comparing Etoposide with Etoposide plus MG132 (supplemental figure S4A). However, this does not answer the question, as p53 is supposed to be stabilized anyway when DNA damage is present. If anything, they should compare MG132 without Etoposide to Etoposide without MG132. That should then be equal. However, even this would need further experimental back up.

Furthermore, they also use cycloheximide and show that they find less p53 protein. However, they did not include the control of cells without Etoposide or Actinomycin D, but with cycloheximide. And even if this would fit the theory, more evidence would be necessary (as total inhibition of protein synthesis is rather crude). Ample evidence would be needed that the specific TIA1-Trp53 mRNA interaction explains this effect.

In fact, experiments with the Trp53 3' UTR suggest that the translational effect is rather limited, as a less than 2-fold different was observed in the luciferase experiment (figure 3K and 3L), whereas protein levels seem to change around 10-fold. Therefore, the authors should also directly study the effect of protein degradation regulation.

Furthermore, the authors should also explain the presence of a sizeable fraction of Trp53 mRNA in the (large) polysome fraction even in the absence of DNA damage, so with very little p53 protein present in the cells (e.g. figure 3F and 4E). Doesn't all of this point more towards constant Trp53 mRNA translation, but quick degradation of the protein?

Without a good answer to these questions, I do not see a good reason to reconsider the model in which p53 is regulated at the level of protein degradation, rather than translation.

Point-by-point response to reviewers

We thank all reviewers for their insightful comments and hope that our responses have improved the manuscript so that it is now worthy of publication.

Reviewer #1 (Remarks to the Author):

Diaz-Munoz and colleagues study Tia1, a translational silencer that can localize to stress granules (SGs), in post-transcriptional regulation of the DNA damage response during B cell activation. Studies of mRNP granules are important in post-transcriptionally regulating mRNAs but are not new; what is new here is the context during B cell activation and DNA damage, as can occur physiologically by DNA replication, CSR, and SHM in activated B cells. The authors argue that mRNA translation of pre-existing mRNAs is a key regulatory mechanism in B cell activation, which is at least partly controlled through the sequestration of Tia1-bound mRNA targets in SGs in an ATM-dependent release mechanism. They provide evidence that Tia1 binds to the 3'UTR of p53 mRNA to sequester it in SGs and suppress p53 mRNA translation. The release of p53 from SGs depended on activation of ATM kinase sensor of the DNA damage response. A list of Tia1 mRNA targets is identified in activated B cells, which may be part of an unappreciated program that regulates B cell development through translational repression and release with DNA damage, such as from CSR/SHM. The study reports an interesting mechanism for the rapid translation of DNA damage modulators in activated B cells, which provides context for SGs beyond model systems. The story is appealing but physiological relevance to implicate mRNA-Tia1 SG sequestration and release with translation as a significant mechanism regulating B cell development and differentiation appears incomplete at this stage and additional issues to be addressed arose during review including:

Response – We thank the reviewer for his helpful insight into our manuscript. Our study provides the first integrated analysis of transcriptomic, translational and RNA protein interactions in a non-transformed model of DNA damage and describes a novel mechanism regulated by Tia1 that controls protein synthesis in primary B cells prior and after DNA damage induction. More importantly, it sets the ground for future in vivo experiments to reveal the importance of mRNA-Tia1 SG sequestration and release in B cell physiology. We agree with the reviewer that much more needs to be done and we are now generating B-cell conditional knock out mice to test how Tia1 and Tial1 control mRNA translation in B cells during Ig recombination, class-switch recombination and somatic hypermutation. We discuss this further in response to reviewer 1, point 7.

1. Page 4 text and Figure S1F- The formation of Tia1 containing SGs in activated B cells suggested that translational silencing occurs during B cell activation based on co-localization with eukaryotic initiation factor eIF3 η . However, not all Tia1-positive SGs co-localize with eIF3 η , and complex co-localization does not impute function. Has the association of SGs with polysomes for translation been examined or excluded in activated B cells?

Response – We agree that colocalization does not imply a functional relationship, but we are not aware of evidence that polysomes are associated with stress granules. The consensus view is that mRNAs, potentially associated with the small ribosomal subunit but not with ribosomes, and prion-like proteins like Tia1 and G3BP participate in SG assembly (Nucleic Acids Res. 2014 Jul;42(13):8678-91.; J Cell Sci. 2009 Oct 15;122(Pt 20):3619-26; Mol Biol Cell. 2004 Dec;15(12):5383-98.; Mol Biol Cell. 2002 Jan;13(1):195-210.). Therefore, we have examined the distribution in sucrose gradients of Tia1, G3BP1, eIF3 η and PABP1 (**new supplementary figure 3, discussed in page 6, line 131-132**) in activated B cells. Tia1 and G3BP1 are only found in ribosome-free fractions (being absent from polysome and monosome fractions), reinforcing the idea that Tia1-dependent translation silencing of targeted mRNAs occurs in SGs where both proteins accumulate. Similarly, eIF3 η is mostly found in ribosome-free fractions, although a small fraction of eIF3 η can be detected associated with monosomes and polysomes likely relating to its functions during translation initiation. PABP1 distribution across polysomes was analysed as an internal experimental control. PABP1 is not only involved in translation initiation but in recycling of terminating ribosomes and it is distributed in both monosome- and polysome-containing fractions.

Etoposide inhibited global mRNA translation in B cells, but not p53 translation. This is in line with previous observations in MCF-7 cells that suggests that p53 translation following DNA damage is initiated in a CAP-independent manner (Oncogene. 2006 Aug 3;25(33):4613-9.). Indeed, ribosome footprinting (Nat Immunol. 2015 Apr;16(4):415-25.; Nat Protoc. 2012 Jul 26;7(8):1534-50.) shows binding to the p53 IRES sequence by ribosomes in LPS-activated B cells (**new figure 4F, discussed in page 9, lines 235-238**). Moreover, detailed analysis of p53 protein synthesis upon DNA damage indicates that only global inhibitors of translation (cycloheximide and puromycin), but not the eIF4E/eIF4G-specific cap-dependent translation initiation inhibitor 4EGI-1 (Cell. 2007 Jan 26;128(2):257-67), block p53 protein synthesis (**new figures 4D and 4E, discussed in page 9, lines 230-235**).

2. Figures 3E and S3C, S3D- p53 mRNA did not increase with B cell activation and massive DNA damage with etoposide, although the heavy polysome associated p53 mRNA fraction increased. Gadd45a and Cdkn1a, known p53-dependent target genes, were induced. Was Gadd45a and Cdkn1a gene induction blocked with sh-p53 or other methods that impair p53 translation, stabilization, or nuclear localization? The data seems incomplete to impute increased p53 translation in activating these genes without showing increased p53.

Response – Our data shows that p53 mRNA translation, but not p53 mRNA abundance, is increased upon induction of DNA damage with etoposide. This results in increased p53 protein expression and activation, as shown by phosphorylation of p53 in Ser15 (**figure 4 and figure 5**). In order to understand whether Gadd45a and Cdkn1a gene induction upon DNA damage was dependent on p53 activation, we pre-treated LPS-activated B cells with the p53 inhibitor pifithrin- α (PFT α) (Science. 1999 Sep 10;285(5434):1733-7.; J Biol Chem. 2004 Jul 16;279(29):30195-201.) one-hour prior induction of DNA damage with etoposide (**new supplementary figure 4E and 4F, discussed in page**

7, lines 182-186). Analysis of Gadd45a and Cdkn1a mRNA and protein levels after 4 hours showed that PFT α prevents Gadd45a mRNA up-regulation in a dose-dependent manner. By contrast, Cdkn1a mRNA induction by etoposide was not affected by B cell pre-treatment with PFT α although p21 protein expression was reduced. Taken together, these new data suggest that Gadd45a induction by etoposide in B cells is partially dependent on p53 activation. Gadd45a and Cdkn1a regulation following DNA damage is more complex as growing evidences suggest that both transcription and post-transcriptional regulation control Gadd45a and p21 expression (Mol Cell. 2010 Oct 8;40(1):34-49.; Cell Cycle. 2011 Jan 1;10(1):23-7. Front Genet. 2012 Aug 25;3:159.; PLoS Genet. 2016 Sep 8;12(9):e1006306.).

3. Figure 3H- p53 co-localizes with Tia1-positive SGs in activated B cells and DNA damage with etoposide releases p53 from Tia1. However, confocal microscopy does not convincingly support this claim. Significant overlap between p53 mRNA and Tia1 protein appears to remain after etoposide treatment. Additionally, accumulation of Tia1 and p53 mRNA in SGs should be quantified, image resolution should be improved, and axis labels should be added on the fluorescence intensity 3D maps to clarify the data.

Response – We have now improved the quality and magnification of the images as well as adding axis labels to the fluorescence intensity 3D maps (**new figure 3H, supplementary figure 5A**). We did not pursue quantitation as the coincident immunofluorescence of Tia1 and p53mRNA cannot be argued to be indicative of a physical association. Instead we focussed on RNA immunoprecipitation experiments to quantify Tia1-p53 mRNA interaction and iCLIP experiments to identify Tia1 binding sites to a single nucleotide level of resolution (**figure 1 and supplementary figure 2**). iCLIP was also attempted in B cells following DNA damage with etoposide, but these Tia1 iCLIP libraries contained extensive read duplication following PCR amplification and were not informative. This likely results from the diminished RNA binding of Tia1 following treatment with etoposide and is consistent with our RNA immunoprecipitation assays that show that Tia1 dissociates from its mRNA targets following DNA damage (**figure 5D and others**).

4. Figures 3I, S3G- are these replicates of the same experiment? Is the Figure S3G right panel a replicate experiment of Figure S3F?

Response – Data shown in **figures 3I and supplementary figure 5F** (*old figure S3G*) are from two independent experiments (performed on 08/05/2010 and 31/06/2010) in which the number of experimental replicates varied from one to three (LPS = 3 and 2 replicates; LPS+Etoposide = 2 and 1 replicates). **Supplementary figure 5D** shows data from one experiment (performed on 31/06/2010) with three experimental replicates per condition. These results were confirmed in subsequent experiments. Tia1-p53 mRNA co-immunoprecipitation in LPS-activated B cells was tested in eight independent experiments and pooled data is shown in **figure 5D**. Tia1-p53 mRNA co-

immunoprecipitation in LPS-activated B cells was analysed in three independent experiments and pooled data is shown in **supplementary figure 6G**.

We have now modified all these figures to show individual data points, means and statistical analysis. Figure legends have been modified accordingly to state the number of independent experiments, the number of experimental replicates and the statistical test performed (Mann-Whitney non-parametric test). A checklist containing information about reproducibility and statistics is now provided for quick reference.

5. Figure 3K, 3L- while statistically significant, the luciferase reporter repression assays from the p53 3'UTR with or without the Tia1 binding site are quite modest at <2-fold up to 3-fold changes. Combined with Figure S3C inability to show changes in p53 protein levels with activation and more physiologic levels of DNA damage from CSR/SHM, this does not give much confidence that this is a major regulatory mechanism for B cell development with activation and DNA damage response.

Response – The luciferase reporter assays were performed in HEK293 cells to functionally validate Tia1-dependent p53 mRNA silencing as well as to evaluate whether distal RNA regulatory elements, to which Tia1 associates, were required to induce mRNA translation upon etoposide treatment. We agree with the reviewer that the changes reported in these experiments are modest, but they show that p53 mRNA translation is subjected to post-transcriptional regulation and the HEK293 system allows us to dissect the nucleotide sequences within the 3'UTR that mediate this effect. Indeed, our results are in line with previous data showing the importance of U-rich elements for p53 mRNA stability and translation (Anticancer Res. 2008 Sep-Oct;28(5A):2553-9). It is now becoming clear that several RNA binding proteins, including Tia1 and HuR, as well as miRNAs can control p53 protein expression post-transcriptionally in a cell-type specific manner (Biochem J. 2016 Oct 15;473(20):3597-3610; Mol Cell Biol. 2015 Apr;35(8):1329-40.; Anticancer Res. 2008 Sep-Oct;28(5A):2553-9; Proc Natl Acad Sci U S A. 2003 Jul 8;100(14):8354-9). Therefore, we focused our efforts on understanding the molecular mechanisms that control p53 expression in primary B cells. In this revised version of the manuscript we have extended our analysis to rule out transcriptional and proteasome-mediated regulation of p53 expression in B cells (**new figure 4 and supplementary figure 6A, discussed in page 8-9, lines 211-241**). We also cite some of the papers highlighted above. This facilitates readers to follow up relevant literature.

6. Figure S4A- are the p53 protein level changes statistically significant to make the claim that CHX blocked p53 translation from pre-existing mRNAs with etoposide or actinomycin-D?

Response – We have included additional data to show the importance of p53 mRNA translation for protein synthesis following DNA damage. In addition to cycloheximide, we have now used puromycin to block global mRNA translation and 4EGI-1 to block CAP-dependent translation (4EGI-1 is a specific inhibitor of eIF4E/eIF4G association, Cell. 2007 Jan 26;128(2):257-67). Data from two

independent experiments is shown in **new figure 4D and 4E, discussed in page 9, lines 230-235**. Statistics (two-way ANOVA and Bonferroni's multiple comparisons test) are described in the figure legend.

7. Text lines 215, 216- the claim is that activated B cells translate p53 from pre-existing mRNA pools blocked by Tia1 in SGs once DNA damage has caused the dissociation to enable cell cycle arrest, DNA damage repair or apoptosis. However, the DNA damage induced in this study is well beyond physiologic levels with targeted CSR and SHM limited mainly to IG genes, and B cells activated by LPS or CD40/IL4/IL5, undergo CSR/SHM and cycle rapidly while dealing with limited IG locus repair. Therefore, it is not clear whether this mechanism is physiologically relevant during B cell activation. It is a key point to address whether loss of Tia1 function in activated B cells can induce premature p53 activation, cell cycle arrest and whether gain of Tia function can inhibit DNA damage repair and apoptosis. Additionally, the requirement of Tia1 in promoting efficient CSR and plasma cell differentiation is also key to address to determine the physiological relevance of Tia1 during later stages of B cell development focused on here.

Response – We agree with the referee and we now clearly state the differences in magnitude between etoposide-induced DNA damage and physiologic levels of DNA damage occurring in B cells as a consequence of CSR and SHM (**page 14, lines 357-360**). We also agree on the importance of validating the mechanism described in our paper in a more physiological set up. We are now generating Tia1 and Tial1 conditional mice to selectively knock down these proteins at different stages of B cell development where VDJ recombination, CSR and SHM take place. AID-induced genotoxic stress in GC B cells is tightly linked to B cell proliferation, selection and terminal differentiation. These are highly complex and regulated processes that will require careful evaluation using in vitro and in vivo approaches to elucidate how Tia1- mediated control of mRNA translation shapes the germinal centre and the antibody response. These studies will take several years to complete and we would like to generate a substantial body of data using these mice before publication.

8. In Figure 4D, the interaction between Tia1 and p53 mRNA is suppressed by ATM kinase activation during the DNA damage response. Based on this model inhibition of ATM kinase after induction of DNA damage should restore the physical interaction between Tia1 and p53 mRNA. There is no statistical significance for data presented in Figure 4D which raises the question whether the interaction between Tia1 and p53 mRNA is actually ATM kinase dependent.

Response – As indicated previously we have now modified **figure 4D and supplementary figure 6G** to show pooled data from eight and three independent experiments respectively. One to three experimental replicates were performed in each of the experiments. All individual data points and mean is shown in the new figure. A Mann-Whitney test has been performed to show that Tia1-p53 mRNA dissociation following DNA damage is ATM kinase dependent ($p=0.0314$).

9. The claim that Tia1 can globally control translation in response to DNA damage seems an overreach as the inhibition of ATM kinase did not suppress translation/polysome accumulation of all Tia1-mRNA targets after induction of DNA damage. There appeared to be no effect on H2-K1 translation after ATM inhibition in Figure S5D, which was identified as a top Tia1 mRNA target in Figure 3D.

Response – We have now revised our claims in the manuscript carefully. Our data shows that Tia1 can widely control translation of hundreds of mRNAs, but the extension of these changes needs to be carefully analysed in a gene-specific manner. We focus our attention on understanding how Trp53 is regulated due to its well established implication in cell physiology and pathology. Following DNA damage with etoposide a number of kinases are activated including ATM, ATR and DNA-PKs. Recent reports have shown that phospho- H2A.X foci formed following etoposide treatment are still present in ATM KO cells (Nat Commun. 2014 Feb 27; 5:3347.). Also it is known that ATM, ATR and DNA-PKs have non-redundant functions (Genes Dev. 2003 Mar 1;17(5):615-28.; Cold Spring Harb Perspect Biol. 2013 Sep 1;5(9).; Cell Rep. 2015 Nov 24;13(8):1598-609.). Therefore it is not beyond expectations that many transcripts are regulated by other means (**discussed in pages 13-14**).

Reviewer #2 (Remarks to the Author):

Regulation of mRNA translation and subcellular location controls protein synthesis of key modulators of the DNA damage response during B cell activation

Diaz-Munoz MD et al

Lymphocyte development into antibody producing cells involves programmed DNA damages for generation of diverse B cell receptors. This manuscript addresses the roles of stress granules and translational control in B lymphocyte activation and DNA damage response. Using CLIP, RNA-seq and Ribo-seq analyses, this manuscript suggests that in response to B cell activation with mitogens, there is formation of RNA stress granules that are critical for translation repression of a large collection of mRNAs with shared U-tracts in their 3'-UTR. Importantly, DNA damage triggers dissociation of p53 mRNA from Tia1 protein in the RNA granules, facilitating robust ribosome translation of p53 mRNA. Increased expression of p53 would then direct genetic programs that feature cell cycle arrest, DNA damage repair, and regulation of apoptosis. Release of p53 from stress granules involves ATM phosphorylation of p53 by mechanisms not yet resolved. Release of the p53 from stress granules was shown to involve the 3'-UTR portion of the p53 transcription. Robust translation of p53 mRNA released from stress granules during B cell development and DNA repair is suggested to involve an IRES embedded within the 5'-portion of the p53 mRNA, although this idea was not tested in the manuscript.

Response – We have performed further experiments to show that p53 is translated in B cells in a CAP-independent manner following DNA damage. We show ribosome association with the p53 IRES in B cells (**new figure 4F, discussed in page 9, lines 235-238**). Furthermore, inhibition of global and CAP-dependent translation with cycloheximide, puromycin or 4EGI-1 inhibitors shows that p53 is translated in B cells in a CAP-independent manner following DNA damage (**new figure 4D and 4E and supplementary figure 6A, discussed in page 9, lines 230-235**).

Overall, this manuscript addresses significant ideas concerning the roles of translational control in lymphocyte development. The central role of p53 trafficking to and from RNA granules during this process are novel, and the manuscript features large scale analyses of mRNAs localized to stress granules and their mRNA and translation changes during the DNA damage response. The Introduction of the manuscript is clearly written, and the experiments presented generally flow logically. The primary concern with the manuscript is that there are some conclusions that are not clearly drawn from experiments. Furthermore, there is uncertainty about some of the quantitation presented in the Results. Finally, the manuscript would be bolstered by testing key features of the p53 model. Clarifying these experimental results and interpretation and testing key features of the p53 model are important to support the stated conclusions.

Response –We have revised the manuscript to more clearly communicate conclusions drawn from the experiments. Also, we have performed new experiments to test key features of the p53 model as suggested. The role of Mdm2 and the proteasome are well recognized as regulators of p53 protein levels in eukaryotic cells. Therefore, we have now investigated whether protein degradation controls p53 expression in activated B cells (**new figure 4A to 4C, discussed in page 9, lines 221-229**). The new results reinforce our conclusion that modulation of mRNA translation controls p53 expression in B cells.

Reviewer concerns:

1. Figure 2, line 133: This section discusses a polysome/RNA-Seq analysis and highlights the importance of translational control in the activation of B cell undergoing a DNA damage response. There is a discussion of gene numbers and percent of those that are differentially expressed upon etoposide treatment that is difficult to follow. The reasoning and the justification for the conclusions are not clear to the reader. It is stated that there are 6340 gene differentially expressed. The term differentially expressed is very general and infers those with mRNA and/or translation changes involving either repression or activation. It is stated that "53% of these genes showed no significant changes in the amount of mRNA associated with ribosome or their distribution along polysomes (Fig. 2D, blot dots), suggesting that changes in mRNA abundance are not always linked to changes in translation." This sentence is not clear regarding both the reasoning and conclusion. This section continues with the statement that 3580 genes were differentially translated upon DNA damage induction (Fig. 2B and supplementary Table 3)." If 53% of the 6340 genes that were significantly measured showed no changes in the amount of mRNA associated with ribosomes, how would 3850 genes (~61%) be differentially translated? This section continues and states that "83% of these genes were differentially expressed, linking their cellular mRNA abundance with translation." This sentence is not clear regarding the meaning or quantitation.

Response – We apologise for the lack of clarity and we have revised the manuscript to convey the results with greater clarity. We now indicate in the results section (page 6, line 142-143) and in the methods section (page 24, lines 637-640) that Ensembl gene annotation was used as 3' end RNA sequencing is unable to identify and quantify all alternatively spliced mRNA transcripts. Transcripts encoded by 6340 genes changed in mRNA abundance upon etoposide treatment. From these 6340 genes, 2985 also changed in the amount of mRNA associated with ribosomes (aprox. 47%), the other 53% of genes (3355 genes) being unchanged. The 3580 is the sum of 2985 genes (which abundance and ribosome association changed upon treatment) and 595 genes (with unchanged mRNA abundance but with a significant change in ribosome association).

The final conclusion at line 153 is also problematic: "Taken together, these results show that the expression of a significant fraction of genes in activated B cells is independent of the overall cellular mRNA content and relies on post-transcription regulation of mRNA translation." It is not

clear how one can conclude that the overall cellular mRNA content is not critical to the activation of B cells.

Response – We agree with the reviewer that overall mRNA content is important for B cell activation and did not intend to conclude otherwise. We have attempted to convey our conclusion, that there is not always a correlation between mRNA abundance and translation, with greater precision in this revised manuscript. Post-transcriptional regulation of mRNA translation by Tia1 is, in part, responsible for this lack of correlation affecting hundreds of transcripts. We have focused our attention on describing how Tia1-binding silences p53 expression in B cells. We agree that further validation is required, likely in a transcript-dependent manner, due to the extensive genome-wide changes observed following DNA damage. In this manuscript we set up the ground to do so, as we do not only provide a comprehensive list of mRNA targets that are potentially regulated by Tia1 but we identify Tia1-RNA interactions to a single nucleotide level.

2. Figure 2, line 156 concerns the fate of the Tia1 associated transcripts. This section begins with the statement that "77% of Tia1 mRNA targets were differentially expressed, differentially translated or both following etoposide treatment. One typically views gene expression to involve transcription, mRNA decay, mRNA translation, and proteolysis, which collectively determine the level of protein expressed from a given gene. Furthermore, the emphasis here is on differential repression/activation of gene expression that occurs with mRNA trafficking to RNA granules, and one would argue this presentation should be more clearly delineate those with lowered translation or reduced mRNA levels from those being induced. Fig. 2G could be explained more clearly, and it is not clear how 20% of Tia1 mRNA targets were increased upon etoposide treatment from the data presented in Fig. 2H. In this case and many others, there is succinct conclusions drawn and quantitation without adequate reasoning that is discussed in the text. It is acknowledged that journal formats can limit word counts, but clarity of reasoning is imperative.

Response – We have now revised this section to provide greater clarity (see pages 6-7). We include supplementary tables 2 and 3 that contain information about changes in mRNA abundance and ribosome association for all genes expressed by B cells. Supplementary table 1 contains information about all Tia1 mRNA targets detected in B cells by iCLIP. Supplementary table 5 contains information about high-confidence Tia1 mRNA targets with changes in ribosome-associated mRNA translation. These tables are called out (page 6-7, lines 140, 151 and 167).

We would like to clarify that our data do not allow us to infer a direct correlation between the repression or activation of gene expression and mRNA trafficking to or from RNA granules. As we state in the results section of the manuscript (page 5, lines 116-117 and page 6, lines 131-132), it is possible that not all transcripts bound by Tia1 are located in SGs. Our approach of integrating Tia1 iCLIP, mRNAseq and PolyRibo-Seq has revealed that some mRNAs (i.e. p53 mRNA) are subject to post-transcriptional silencing in B cells, but a large amount of further work is required to analyse the mRNA composition of SG prior to and after induction of DNA damage. As we have indicated in our

discussion (**page 12, lines 313-315**), further characterization of the RNA:protein interactions of other key components of SGs will increase our understanding of how these mRNAs are regulated within these highly dynamic entities. We were unsuccessful making Tia1 iCLIP libraries from B cells treated with etoposide. This is not surprising as we found that Tia1 dissociates from its mRNA targets following DNA damage.

Finally, we include new evidence that establishes a correlation between those 20% of Tia1 mRNA targets (such as p53) that are increased following DNA damage and CAP-independent translation. For p53 in particular, ribosome footprints show that the p53 IRES is occupied by ribosomes (**new figure 4F, discussed in page 9, lines 235-238**). p53 mRNA translation is unaffected by inhibition of CAP-dependent translation with 4EGI-1. Only global inhibition of mRNA translation with CHX or puromycin blocks p53 protein synthesis induced by etoposide (**new figure 4D, 4E and supplementary figure 6A, discussed in page 9, lines 230-235**).

3. Figures 3 and 4: What are the numbers 1-4 representing at the bottom of Fig. 3D? Fig. 3M was not fully clear regarding the experiment arrangement. In Fig. 3F and other polysome profiling experiments that follow, the described normalization (18S rRNA and beta2m transcript) does not appear to be appropriate. For example, in Fig. 4E there appears to be an increase in polysome association of p53 mRNA upon etoposide treatment that is lowered by ATM inhibitor. However, the pattern of mRNA distribution does not show a shift in mRNA from polysomes to lighter fractions, rather just lowered transcript levels, which is inconsistent with experiments presented in the manuscript. This reinforces the importance of appropriate normalization (e.g. spike in controls or others) in the polysome profiling.

Response – Numbers in figure 3D indicates biological replicates (figure legends have been reviewed). Data from polysome profiling + sequencing experiments (**figure 3F, 5E and supplementary figure 7C and 7D**) was not normalized by the means indicated by the reviewer (18S rRNA and B2m transcript). Data is shown as cDNA counts (number of reads mapped to a given transcript in each fraction after DESeq2 data adjustment to a negative binomial distribution and normalization). Validation was performed by polysome profiling by qPCR and normalised by Hrpt (**supplementary figure 4D**).

Polysome profiling shows that in LPS-activated B cells a small fraction of p53 mRNAs are associated to several ribosomes and settle within high sucrose-density fractions. New ribosome footprinting data (**new figure 4F, discussed in page 9, lines 235-238**) indicates that ribosomes are unevenly distributed along the p53 transcript. Ribosome footprints accumulate in the 5'UTR that contains the p53 IRES. This suggests a tight control of translation initiation, although this point is difficult to validate in primary B cells and remains unresolved. Nevertheless, several reports have now suggested that p53 mRNA translation can be regulated at both initiation and elongation (EMBO Rep. 2006 Apr; 7(4): 404–410.; Cancer Res 2009; 69: (22). November 15, 2009; Mol Cell Biol. 2015 Dec; 35(23): 4006–4017.). We have been unable to detect p53 protein by Western blot using protein lysates from LPS-activated B cells, even in the present of proteasome, Mdm2 or Usp7 inhibitors. These results are now presented and discussed in **pages 9, lines 219-229**.

4. The underlying model argues that key sequences in the 3'-UTR of p53 transcript allows for association and controlled release of this transcript from the RNA granules during the DNA damage response. There is speculation that there is competition between Tia1 and one or more RNA binding proteins that associate with p53 and facilitate this release and ATM is involved in this process. How would ATM, and possibly p53 protein phosphorylation, fit into this model?

Response – Although it is well known how ATM kinase contributes to p53 phosphorylation and activation following DNA damage (J Biol Chem. 2002 Apr 12;277(15):12491-4.; Nat Rev Cancer. 2009 Oct;9(10):714-23.), much less is known about how ATM controls protein-RNA binding specificities and affinities in vivo. Our data highlights for the first time that ATM activation promotes mRNA translation of key modulator of the DNA damage response like p53 by controlling Tia1 association with U-rich elements present in the 3'UTRs of its mRNA targets.

Tia1 competes with other RNA binding proteins like HuR (iCLIP data in Nat Immunol. 2015 Apr;16(4):415-25) to bind to these U-rich elements. Previous reports indicate that both HuR and Tia1/Tial1 can be phosphorylated upon induction of DNA damage and apoptosis (J Exp Med. 1995 Sep 1;182(3):865-74.; Mol Cell. 2010 Oct 8;40(1):34-49.; Mol. Cell 2007;4:543-557.), although the consequences of such post-translational modification remain unclear. In-silico analysis has predicted the existence of two Atm-specific phosphorylation sites conserved in the second RNA-recognition motif of Tia1 and Tial1 (S209/S210 in Tia1; S217/S218 in Tial1), which importance is currently under investigation. We are now generating HuR, Tia1 and Tial1 conditional mice to investigate the role of these proteins in B cells undergoing CSR and SHM.

How would omission of the proposed Tia1 binding site in the p53 mRNA block trafficking to RNA granules and lead to elevated p53 mRNA translation even in the absence of the DNA damage response? What consequence would this have on the B cell processes?

Response – It is unlikely that deletion of U-rich elements in the p53 3'UTR reveals how p53 mRNA cellular location and translation is regulated in B cells as it is unknown the full list of RNA binding proteins and ncRNAs involved in p53 mRNA regulation. Previous reports have highlighted the importance of the U-rich elements present in the 3'UTR for p53 mRNA stabilization and translation (Biochem J. 2016 Oct 15;473(20):3597-3610; Mol Cell Biol. 2015 Apr;35(8):1329-40.; Anticancer Res. 2008 Sep-Oct;28(5A):2553-9; Proc Natl Acad Sci U S A. 2003 Jul 8;100(14):8354-9). In this study we have performed luciferase reporter assays in HEK293 cells to functionally validate Tia1-dependent p53 mRNA silencing. But RNA binding proteins like HuR, PTB and nucleolin can bind and compete with Tia1 for the same binding motifs. HuR binding to p53 U-rich elements promotes p53 mRNA translation (Proc Natl Acad Sci U S A. 2003 Jul 8; 100(14): 8354–8359.). HuR competes with Tia1 and ncRNAs that silence p53 translation (Nucleic Acids Res. 2014 Sep;42(15):10099-111.).

Some test of the key tenets of the model and its consequences on B cell fate would strengthen this manuscript.

Response – One of the main objectives of this study is to understand how Tia1-dependent regulation of mRNA translation controls p53 expression in primary B cells and how integrates with other well described mechanisms for p53 regulation. In this new version we have deeply investigate how proteasome-mediated regulation control p53 expression to conclude that protein-degradation has a minor role in LPS-activated B cells (**new figure 4, discussed in page 9 and page 14**).

Minor:

The abstract mentions protein translation, which should be referred to as mRNA translation or protein synthesis.

Response – This is now changed to protein synthesis.

Reviewer #3 (Remarks to the Author):

The authors describe a study to identify mRNAs that interact with TIA1 in stress granules and may therefore be subject to regulation of translation. They follow up primarily on Trp53. They argue that p53 mRNA is highly expressed in developing B lymphocytes, but the mRNA is repressed by interaction with TIA1. This binding is released upon DNA damage induction, followed by translation.

The authors do a good job analyzing the stress granule associated mRNAs and changes in abundance of mRNAs in general and in the polysome fraction upon B lymphocyte differentiation and DNA damage induction. However, the part on regulation of mRNA expression at the level of translation is less clear. The model is very attractive: Trp53 mRNA is sequestered and only translated when needed. However, the data do not fully support this and are definitely not conclusive, as outlined below. It is an interesting idea, but will need much better experimental data before it can be considered a credible new mechanism of Trp53 regulation.

Response – We thank the reviewer for the critique and suggestions. We have now performed further experiments to evaluate the role of previously described mechanisms for p53 regulation in the context of activated B cells. In summary, these new data reinforce our conclusion that p53 mRNA translational control is the major regulatory mechanism for p53 expression in primary B cells, having proteasome-dependent degradation a minor effect if any.

The widely accepted model for p53 protein regulation mainly considers degradation via hdm2 (or mdm2 in the mouse) when no DNA damage is present. Upon DNA damage induction, p53 is phosphorylated and loses its affinity for hdm2, resulting in protein stabilization, because it is no longer degraded by the proteasome. The authors do very little to investigate this mechanism. The only thing they try is comparing Etoposide with Etoposide plus MG132 (supplemental figure S4A). However, this does not answer the question, as p53 is supposed to be stabilized anyway when DNA damage is present. If anything, they should compare MG132 without Etoposide to Etoposide without MG132. That should than be equal. However, even this would need further experimental back up.

Response –We have now analysed the expression of p53 in B cells treated with MG132 in the absence of etoposide and compare it with the expression of p53 in B cells treated or not with etoposide (**new figure 4A, discussed in pages 9-10, lines 215-220**). Proteasome inhibition with MG132 did not increase p53 protein expression in B cells. A similar result was obtained when the proteasome inhibitor lactacystin was used, suggesting that p53 degradation by the proteasome is not a major regulator of p53 expression in this system. This conclusion was further reinforced by a set of experiments in which specific inhibitors of Mdm2, Mdmx and Usp7 were used (**new figure 4B and 4C, discussed in page 9, lines 221-229, and in page 14, lines 360-362**). None of them was able

to increase p53 protein in B cells. And they also failed to increase p53 levels upon etoposide treatment consistent with mdm2 disassociation from p53 following DNA damage. Altogether, these new evidences suggest that proteasome-dependent regulation does not control p53 expression in LPS-activated B cells.

Furthermore, they also use cycloheximide and show that they find less p53 protein. However, they did not include the control of cells without Etoposide or Actinomycin D, but with cycloheximide. And even if this would fit the theory, more evidence would be necessary (as total inhibition of protein synthesis is rather crude).

Response – We have now analysed further the role of mRNA translation during the DNA damage response induced by etoposide in primary B cells by not only blocking protein synthesis with cycloheximide but by using puromycin and 4EGI-1, a specific inhibitor of eIF4E/eIF4G association (Cell. 2007 Jan 26;128(2):257-67) that blocks CAP-dependent translation (**new figures 4D, 4E and supplementary figure 6A, discussed in page 9, lines 230-238, and in page 13, lines 328-330**). Similar to the results shown originally in our manuscript, global inhibition of translation blocks completely p53 protein synthesis induced by etoposide or actinomycin D. Translation of p53 following DNA damage is CAP-independent as 4EGI-1 was unable to block p53 protein synthesis. This is in line with previous reports (Oncogene. 2006 Aug 3;25(33):4613-9.; Oncogene (2006) 25, 6936–6947.; Mol. Cell. Biol. doi:10.1128/MCB.00365-15).

Ample evidence would be needed that the specific TIA1-Trp53 mRNA interaction explains this effect. In fact, experiments with the Trp53 3' UTR suggest that the translational effect is rather limited, as a less than 2-fold different was observed in the luciferase experiment (figure 3K and 3L), whereas protein levels seem to change around 10-fold. Therefore, the authors should also directly study the effect of protein degradation regulation.

Response – Our luciferase reporter assays were conducted in HEK293 cells to functionally validate Tia1-dependent p53 mRNA silencing. Tia1 binds to U-rich elements present close to the polyadenylation site of p53 which are proven to be important for p53 mRNA stability and translation (Anticancer Res. 2008 Sep-Oct;28(5A):2553-9), mechanisms that are regulated by several RNA binding proteins and microRNAs in a cell-dependent manner (Anticancer Res. 2008 Sep-Oct;28(5A):2553-9, Biochem J. 2016 Oct 15;473(20):3597-3610; Mol Cell Biol. 2015 Apr;35(8):1329-40.; Anticancer Res. 2008 Sep-Oct;28(5A):2553-9; Proc Natl Acad Sci U S A. 2003 Jul 8;100(14):8354-9). Therefore, we focused our efforts to understand whether transcription, translation or proteasome degradation control p53 expression in primary B cells rather than in HEK293 cells.

Furthermore, the authors should also explain the presence of a sizeable fraction of Trp53 mRNA in the (large) polysome fraction even in the absence of DNA damage, so with very little p53 protein present in the cells (e.g. figure 3F and 4E). Doesn't all of this point more towards constant Trp53 mRNA translation, but quick degradation of the protein?

Response – Our new experiments indicate that proteasome-mediated degradation contributes little to controlling p53 protein levels in activated B cells (**new figures 4A, 4B and 4C, discussed in page 9-10, lines 214-229**). Although we cannot detect p53 protein by immunoblot with any of the antibodies tested, we cannot rule out that there is a small amount of p53 protein in LPS-activated B cells. Indeed, analysis of ribosome footprints distribution across Trp53 mRNA suggests that translation might be active (**new figure 4F, discussed in page 9**). More importantly, we now identify ribosome stalling that might limit translation initiation and elongation without decreasing the ribosome content per transcript. The presence of an IRES in the 5'UTR of p53 has been previously reported (WIREs RNA 2014, 5:131–139. doi: 10.1002/wrna.1202, (2005) Molecular Cell, 17 (3), pp. 405-416. Wiley Interdiscip Rev RNA. 2014 Jan-Feb;5(1):131-9. Oncogene. 2006 Aug 3;25(33):4613-9. Epub 2006 Apr 10. EMBO Rep. 2006 Apr;7(4):404-10.). These studies have described the existence of several IRES-transacting factors (ITAFs) that control p53 protein synthesis following DNA damage (Cell Cycle. 2008 Jul 15;7(14):2189-98.; Cell Cycle. 2011 Jan 15;10(2):229-40. RNA Biol. 2011 Jan-Feb;8(1):132-42.). Abnormal expression of these ITAFs or mutation of the p53 IRES is linked with defective p53-mediated responses and tumorigenesis (Genes Dev. 2011 Jul 15;25(14):1528-43.; Mol Cell Biol. 2015 Dec;35(23):4006-17.; Oncogene. 2013 Aug 29;32(35):4148-59.). In summary, p53 protein synthesis is regulated at several levels including translation and protein degradation that we are now just starting to understand.

Without a good answer to these questions, I do not see a good reason to reconsider the model in which p53 is regulated at the level of protein degradation, rather than translation.

Response – We have now provided new experimental observations to address all questions. We thoroughly thank the reviewers for their feedback as it has helped us to significantly improve the manuscript.

Reviewers' Comments:

Reviewer #1 (Remarks to the Author)

Additional studies, controls, better explanations, improved statistical presentation, and more carefully stated interpretations and conclusions have improved the revised manuscript.

Reviewer #2 (Remarks to the Author)

Regulation of mRNA translation and subcellular location controls protein synthesis of key modulators of the DNA damage response during B cell activation
Diaz-Munoz MD et al (Turner M corresponding)

This is a revised manuscript that addresses the roles of stress granules and translational control in B lymphocyte activation and DNA damage response. The manuscript suggests that in response to B cell activation, there is formation of RNA stress granules that features TIA1 protein that binds U-tracts in the 3'-UTRs of target genes, repressing their translation. DNA damage triggers dissociation of p53 mRNA from TIA1 protein in the RNA granules, facilitating robust ribosome translation of p53 mRNA. Increased translational expression of p53 (perhaps through an IRES) would then direct genetic programs that are critical modulators of the DNA damage response. There are some interesting ideas in this manuscript that emphasize the roles of translational control in lymphocyte development. The iCLIP analysis of mRNAs that bind TIA1, presumably in stress granules, is of interest. Extending from these genome-wide analyses, the manuscript focuses on potential p53 engagement with TIA1 and stress granules and the role of this interaction in differential regulation of p53 translation and DNA damage responses. In this regard, the manuscript has rather broad ambitions. There were two central concerns about the manuscript, which were detailed in the earlier review. First, many of the experiments were not clearly interpreted or did not sufficiently support the stated conclusions. The revision addressed many of these concerns, but some remain, especially regarding the model of p53 translational control. Second, the manuscript would be bolstered by testing key features of the p53 model stated in the abstract. This concern was largely not addressed, and the added experiments, such as that involving proteasome inhibition, did not test key features of the proposed model.

Reviewer concerns:

1. There are still some concerns about clarity of text and experimental results. An example of text in the introduction that is not clear is the following: "Tia1 knockout (KO) mice are embryonic lethal, whereas 50% of Tia1-KO mice die between E16.5 and 3 weeks." What does this mean? The other half is lethal before E16.5? Figure 2 D and E are critical experiments analyzing that used RNA-seq to measure gene transcripts in different polysome fractions, with an eye towards assessing translational control. The entire description of the experiment in the results section is the following "Differential ribosome association highly correlated with changes in polysome distribution (Fig. 2 D and 2E)." The y-axis of both figures is labeled- % of genes with changes in polysome distribution. The x-axis indicates specific fractions. First there is "any fraction." Does this imply the total measurement of a given gene transcript? Next there is the "monosome" fraction. What precisely is the % of genes with changes in the polysome distribution that are attributed to the monosome fraction? In Fig. 2E there is reference to a "RiboLoad" that is "defined as the proportion of reads in heavy polysomes divided by the total of reads mapped to a given mRNA transcript." What does the Differential RiboLoad value in the monosome fraction represent?

2. Supplement Fig. 4D suggests that total p53 mRNA were unchanged upon etoposide treatment, although there was a trend toward higher levels in LPS+Et. However, p53 mRNA association with polysomes increased upon this stress, suggestive of enhanced translation. The levels of transcripts in the LPS versus LPS+Et polysomes follow similar patterns, just with lower amounts of p53 mRNA

across the board in the LPS alone. How would this figure look if plotted as a percent of the total p53 transcript in each fraction for the LPS versus LPS+Et? Differences between the polysome distribution for the p53 mRNA in the polysome fractions in Fig. 3F showed perhaps some changes in levels of p53 transcript fractions 13 and 14, which were indicated to be significant. However, there were no apparent changes in the distribution of p53 transcript levels (say more now found in monosomes are disomes upon translation repression), which would be expected with translational control. The same observation occurred with p53 mRNA levels in polysomes in Fig. 5E. Is the thought here that the mRNA is found in the stress granule and therefore the amount of mRNA available for translation is less? Translation efficiency occurs uniformly when mRNA is released from the stress granule?

3. Figure 3K and L incorporate a luciferase reporter with the 3'-UTR from p53. There was a some reduction of the reporter with overexpression of TIA1 protein. How would this change with in wild-type or TIA1 depleted cells in the presence or absence of etoposide? Results in Fig. 3L are stated to demonstrate p53 translational control. Do the reporter mRNA levels change? Does this depend on the stated TIA1 binding site in the 3'-UTR? Some test of this idea was requested in the original review, and is still needed. There is some discussion in the manuscript that HuR may bind to the same 3'-UTR site and stabilize p53 mRNA or enhance its translation. This was the underlying reason for why the manuscript did not test the idea that omission of the TIA1 binding site in the p53 mRNA would alter the proposed translation and accompanying expression of target genes in the DNA damage response. It could be argued that if TIA1 is missing, then the competition between these RNA binding proteins for the p53 mRNA is moot. The proteasome inhibition experiments did not shed additional light on this proposed model.

4. The experiments in Fig. 4E and F would need to be bolstered to conclude that an cap-independent IRES mechanism underlies translational control of p53 in this model system.

Reviewer #3 (Remarks to the Author)

The authors have added quite some new data supporting their claims. This makes the paper stronger and makes some conclusions more solid. However, the point of p53 regulation by translational control does not have a very strong basis in the data. The authors show that just adding MG132 does not result in increased p53 protein levels. However, positive proof of increased translation depends largely on the crude experiment of total inhibition of translation or the absence of an effect of degradation inhibitors.

The experiment in HEK293 cells would point towards a minor role of TIA1 in p53 translation control. The authors argue that this is because the experiment was not done in B cells. Therefore, they should do the effort to show that the situation is different if they would repeat the experiment in B cells and that the p53 UTR is then indeed the translational control element that they argue it is. It is not very convincing to say that TIA1 inhibits translation, when the luciferase expression levels are higher in the full length than with the short version of the 3' UTR (figure 3L). That would more point towards activation of translation by the longer UTR than inhibition by TIA1. This issue needs much better experimental evidence.

Reviewer #1 (Remarks to the Author):

Additional studies, controls, better explanations, improved statistical presentation, and more carefully stated interpretations and conclusions have improved the revised manuscript.

Response – We sincerely appreciate the time and dedication put into refereeing our manuscript by the reviewer and we are grateful for that.

Reviewer #2 (Remarks to the Author):

Regulation of mRNA translation and subcellular location controls protein synthesis of key modulators of the DNA damage response during B cell activation. Diaz-Munoz MD et al (Turner M corresponding)

This is a revised manuscript that addresses the roles of stress granules and translational control in B lymphocyte activation and DNA damage response. The manuscript suggests that in response to B cell activation, there is formation of RNA stress granules that features TIA1 protein that binds U-tracts in the 3'-UTRs of target genes, repressing their translation. DNA damage triggers dissociation of p53 mRNA from TIA1 protein in the RNA granules, facilitating robust ribosome translation of p53 mRNA. Increased translational expression of p53 (perhaps through an IRES) would then direct genetic programs that are critical modulators of the DNA damage response.

There are some interesting ideas in this manuscript that emphasize the roles of translational control in lymphocyte development. The iCLIP analysis of mRNAs that bind TIA1, presumably in stress granules, is of interest. Extending from these genome-wide analyses, the manuscript focuses on potential p53 engagement with TIA1 and stress granules and the role of this interaction in differential regulation of p53 translation and DNA damage responses. In this regards, the manuscript has rather broad ambitions. There were two central concerns about the manuscript, which were detailed in the earlier review. First, many of the experiments were not clearly interpreted or did not sufficiently support the stated conclusions. The revision addressed many of these concerns, but some remain, especially regarding the model of p53 translational control.

Response - We are grateful for the comments, suggestions and concerns raised by the reviewer. Further experiments have been performed to test the model of p53 translational control. Among them, as detailed below, we have performed gain and loss of function experiments that show that Tia1 is indeed an important modulator of p53 translation. We have performed a mutagenesis analysis to evaluate the functionality of Tia1 binding sites in p53 3'UTR. And we show that deletion of Tia1 RNA recognition motifs prevents translational silencing. All together, these new data provide additional strong support for the proposed model.

Second, the manuscript would be bolstered by testing key features of the p53 model stated in the abstract. This concern was largely not addressed, and the added experiments, such as that involving proteasome inhibition, did not test key features of the proposed model.

Response - We have now extensively tested the model of p53 translational control as well as evaluated the role of transcription and protein degradation in p53 protein expression in primary B cells. This was important as transcription, translation and protein stability have each been implicated in p53 expression during the DNA damage response. Although the relative contributions of each of these processes may be different depending on cell type and the nature and context of the DNA damage induced. In summary, our previous findings are supported with new experimental evidence that shows how RNA sequences in the p53 5'UTR and 3'UTR act as regulatory elements to control translation, which is the major driver of p53 protein expression in primary B cells following exposure to etoposide.

Specific reviewer concerns:

1. There are still some concerns about clarity of text and experimental results. An example of text in the introduction that is not clear is the following: "Tial1 knockout (KO) mice are embryonic lethal, whereas 50% of Tia1-KO mice die between E16.5 and 3 weeks." What does this mean? The other half is lethal before E16.5?

Response – Text has now been revised (see page 4, lines 95-97).

Figure 2 D and E are critical experiments analyzing that used RNA-seq to measure gene transcripts in different polysome fractions, with an eye towards assessing translational control. The entire description of the experiment in the results section is the following "Differential ribosome association highly correlated with changes in polysome distribution (Fig. 2 D and 2E)." The y-axis of both figures is labeled- % of genes with changes in polysome distribution. The x-axis indicates specific fractions. First there is "any fraction." Does this imply the total measurement of a given gene transcript? Next there is the "monosome" fraction. What precisely is the % of genes with changes in the polysome distribution that are attributed to the monosome fraction?

Response – We have now revised the manuscript carefully to explain further the experiments shown in figure 2D and 2E (see page 6-7, lines 162-178). We have also reviewed both figures (y-axis have been renamed for clarity) as well as the figure legends. We now carefully explain the meaning of the groups in the x-axis (see page 30, lines 798-807). Briefly, mRNAs were grouped based on significant changes in abundance in any of the 4 to 16 samples (called as "any fraction") or based on significant changes in monosomes (fractions 4 to 7), light polysomes (fractions 8 to 10) or heavy polysomes only (fractions 9 to 16).

In Fig. 2E there is reference to a “RiboLoad” that is “defined as the proportion of reads in heavy polysomes divided by the total of reads mapped to a given mRNA transcript.” What does the Differential RiboLoad value in the monosome fraction represent?

Response – We thank the reviewer for pointing out to Figure 2E as has allowed us to rewrite and bring clarity to this section (see page 6-7, lines 162-178. See page 30, lines 798-807).

Figure 2E shows the likelihood of an mRNA of being differentially expressed or being differentially associated to ribosomes if there is a change in polysome distribution. Briefly, the data show that over 50% of mRNAs are either differentially expressed or associated to ribosomes if the transcript abundance in any of the 4 to 16 fractions is significantly different.

RiboLoad is indeed defined as the proportion of reads in heavy polysomes divided by the total of reads mapped to a given mRNA transcript and it is used as a measure of translation control independent of changes in mRNA abundance. RiboLoad does not change significantly in the conditions tested. This is most likely due to the great changes in both mRNA abundance and ribosome association induced by etoposide. This is now explained in the manuscript (see page 7, lines 171-174).

2. Supplement Fig. 4D suggests that total p53 mRNA were unchanged upon etoposide treatment, although there was a trend toward higher levels in LPS+Et. However, p53 mRNA association with polysomes increased upon this stress, suggestive of enhanced translation. The levels of transcripts in the LPS versus LPS+Et polysomes follow similar patterns, just with lower amounts of p53 mRNA across the board in the LPS alone. How would this figure look if plotted as a percent of the total p53 transcript in each fraction for the LPS versus LPS+Et?

Response – Total p53 mRNA in LPS-activated B cells treated or not with etoposide was measured independently multiple times as part of different experimental approaches (see Fig. 3E, Fig. 7C, Supplementary Fig. 4D and Supplementary Fig. 7C). We did not observe any changes in p53 mRNA in any of the experiments. Therefore, we conclude that p53 mRNA abundance remains constant during the DNA damage response induced by etoposide.

By contrast, the amount of p53 mRNA associated with ribosomes significantly increases upon etoposide treatment (see Fig. 3E). Distribution analysis indicates that indeed the number and the proportion of transcripts in heavy polysomes increases upon etoposide treatment (Fig. 3E, Supplementary Fig. 4D and Supplementary Table 6. Discussed in page 8, lines 200-205). Our data support the notion that this is because a higher number of p53 mRNA transcripts are now bound to ribosomes rather than higher ribosome occupancy per p53 mRNA transcript, that would be reflected

by a profound change in the polysome pattern (e.g. p53 mRNA moving to heavy polysomes from monosomes and/or light polysomes).

The number of cDNA counts and the percentage of mRNA transcripts in each of the fraction is now provided to readers in the new supplementary table 6.

Differences between the polysome distribution for the p53 mRNA in the polysome fractions in Fig. 3F showed perhaps some changes in levels of p53 transcript fractions 13 and 14, which were indicated to be significant. However, there were no apparent changes in the distribution of p53 transcript levels (say more now found in monosomes are disomes upon translation repression), which would be expected with translational control. The same observation occurred with p53 mRNA levels in polysomes in Fig. 5E. Is the thought here that the mRNA is found in the stress granule and therefore the amount of mRNA available for translation is less? Translation efficiency occurs uniformly when mRNA is released from the stress granule?

Response – We now discuss more clearly that the increase in ribosome-associated p53 mRNA upon etoposide treatment (Fig. 3E) is likely to be due to Tia1 dissociation and p53 mRNA release from SGs. To our knowledge, RNA stored in SG is not associated to ribosomes. This is confirmed by the fact that B-cell cytoplasmic extract fractionation using sucrose gradients reveals that Tia1 and other SG-markers like G3BP1 are not detected in ribosome containing fractions (see Supplementary Fig. 3B. Discussed in page 5, lines 135-138). Therefore, any p53 mRNA bound to Tia1 and stored in SG is most likely not been quantified in our polysome fractionation and sequencing analysis (discussed in page 9, lines 239-242). The significant increase in the number and proportion of p53 transcripts in fractions 13 and 14 upon etoposide treatment is likely to be caused by higher mRNA availability due to Tia1 dissociation rather than by a profound change in polysome distribution of p53 mRNA. This suggests that translation efficiency remains constant.

3. Figure 3K and L incorporate a luciferase reporter with the 3'-UTR from p53. There was some reduction of the reporter with overexpression of TIA1 protein. How would this change with wild-type or TIA1 depleted cells in the presence or absence of etoposide?

Response – We have now performed gain and loss of function experiments to reinforce our conclusion that Tia1 controls translation by binding to p53 3'UTR.

Three different dicer-substrate short interfering RNAs have been used to knock down Tia1 expression in HEK293 cells (Supplementary Fig. 5B and 5C). Analysis of luciferase activity of a gene reporter construct with the p53 3'UTR shows that Tia1 knock down significantly increases translation (Fig. 4B. Discussed in page 8, lines 222-226). Importantly, knock down of Hprt (used as experimental control) did not significantly increase translation without affecting mRNA abundance (Supplementary Fig. 5B and 5D. Discussed in page 9, lines 226-227). On the contrary, overexpression

of Tia1 reduces translation of the luciferase reporter. This requires Tia1 binding to the p53 3'UTR as deletion of the RNA recognition motif 1 and 2 rescues translation (Fig 4D. Discussed in page 9, lines 232-236). Taken together, we now provide very strong evidences that Tia1 induces p53 translational silencing.

Results in Fig. 3L are stated to demonstrate p53 translational control. Do the reporter mRNA levels change? Does this depend on the stated TIA1 binding site in the 3'-UTR? Some test of this idea was requested in the original review, and is still needed.

Response – Over expression or knock down of Tia1 did not change reporter mRNA levels suggesting that Tia1 controls translation rather than RNA stability (Supplementary Fig. 5D, 5E, 5F and 5G. Discussed in pages 8-9, lines 222-224, 234-236). Tia1 exerts p53 translational control by binding to two U-rich RNA regulatory elements present in the 3'UTR (Fig 4A and Supplementary Fig. 5A). Mutation of these sequences increases translation of a luciferase reporter gene without changes in mRNA abundance. More importantly, mutation of these sequences rescues translation inhibition caused by Tia1 overexpression (Fig 4D. Discussed in page 9, lines 231-232). The results demonstrate that these sequences function as RNA regulatory elements for p53 translational control and strengthen the proposed model.

There is some discussion in the manuscript that HuR may bind to the same 3'-UTR site and stabilize p53 mRNA or enhance its translation. This was the underlying reason for why the manuscript did not test the idea that omission of the TIA1 binding site in the p53 mRNA would alter the proposed translation and accompanying expression of target genes in the DNA damage response. It could be argued that if TIA1 is missing, then the competition between these RNA binding proteins for the p53 mRNA is moot.

Response – We speculated in the original manuscript about whether Tia1 and HuR compete for binding to the p53 RNA regulatory elements. Although we think this is an attractive possibility, we accept that there is a limited amount of evidence to support the notion. We believe a significant amount of further work would be required to convincingly demonstrate this, and that it would not enlighten the current manuscript, and therefore we have removed the discussion of this hypothesis.

The proteasome inhibition experiments did not shed additional light on this proposed model.

Response – Protein degradation is a major regulatory mechanism to control p53 expression in many cell systems. Thus it needed to be considered as part of a model of how p53 is regulated in response to etoposide in activated mouse B cells. Our data suggest that proteasome dependent regulation of p53 has a minor role in these cells. Thus the importance of the proteasome inhibition

data is that it rules-out p53 protein stabilisation as the dominant mechanism of p53 induction in this system

4. The experiments in Fig. 4E and F would need to be bolstered to conclude that a cap-independent IRES mechanism underlies translational control of p53 in this model system.

Response – We present new data to validate the existence of a functional IRES in p53 5'UTR.

First, we have used ribosome profiling data from B cells treated with cycloheximide to map ribosome footprints to the p53 5'UTR and to identify the IRES nucleotide sequence occupying the ribosome P-site (Fig. 6C, discussed in pages 10-11, lines 286-288). The existence of this IRES in other cells was validated in Ribo-seq libraries from murine bone marrow-derived dendritic cells, mouse embryonic fibroblasts and mouse liver cells treated with the translation initiation inhibitors lactimidomycin and harringtonine (Supplementary Fig. 5H, discussed in page 11, lines 288-290).

Second, we have generated a bicistronic dual luciferase reporter construct to functionally validate the mouse p53 IRES (Fig. 5E and 5F, discussed in page 11, lines 292-303). Mutation of the p53 IRES sequence placed between ORF1 (encoding renilla luciferase) and ORF2 (firefly luciferase) decreased translation of the ORF2 by 5-fold. More importantly, it prevented translation induction after treatment with etoposide without affecting mRNA abundance.

These new data are in line with previous results obtained using general translation inhibitors or after specific inhibition of CAP-dependent translation by blocking eIF4E-eIF4G interaction (Fig 5A and 5B) and provide strong evidence that induction of CAP-independent translation is essential for p53 translation during the DNA damage response of primary mouse B cells.

Reviewer #3 (Remarks to the Author):

The authors have added quite some new data supporting their claims. This makes the paper stronger and makes some conclusions more solid. However, the point of p53 regulation by translational control does not have a very strong basis in the data. The authors show that just adding MG132 does not result in increased p53 protein levels. However, positive proof of increased translation depends largely on the crude experiment of total inhibition of translation or the absence of an effect of degradation inhibitors.

Response – We appreciate very much the reviewer's comments. We have performed new experiments and the data supports our conclusion that post-transcriptional regulation by Tia1 controls p53 mRNA translation in B cells. Neither global inhibition of the proteasome (with MG132) nor specific inhibition of p53 protein interaction with Mdm2 and Mdmx (using RITA, Nutlin3 or SJ172550) suggests that protein degradation plays a major role in p53 expression in primary B cells. Thus, we have focused our study on understanding how gain and loss of Tia1 function modulates translation of p53. The new data indicates that Tia1 binding to two U-rich sequences present in the p53 3'UTR silences translation. Both site-directed mutagenesis of these sequences and overexpression of a Tia1 mutant lacking RNA recognition motifs 1 and 2 prevented Tia1-dependent translational inhibition (New figure 4. Discussed in pages 8-9, lines 215-236). These are strong evidences that Tia1-mediated translational control is at least partially responsible for p53 protein expression in response to DNA damage in activated mouse B cells.

Also, we have performed new experiments to further understand the mechanism that enables increased p53 translation after induction of a DNA damage response. Similar to human, the mouse p53 5'UTR has an internal ribosome entry site (IRES) that is widely used in different cell types including B cells (Figure 6C and new supplementary figure 5H. Discussed in page 11, lines 286-290). Analysis of ribosome footprints mapped to the p53 5'UTR has enabled us to identify the sequence of the p53 IRES for functional characterization. Generation of a bicistronic mRNA luciferase reporter construct, in which translation of a second open reading frame (ORF2) encoding the firefly luciferase protein is regulated by the p53 5'UTR, shows that the p53 IRES is not only functional but it is responsible for an increase in mRNA translation upon induction of DNA damage with etoposide (New figures 6E and 6F. Discussed in page 11, lines 290-303). These new results are consistent with and extend our previous observations using global inhibitors of translation (cycloheximide and puromycin) and specific inhibitors of CAP-dependent translation (eIF4E specific inhibitor 4EGI-1). They highlight the importance of IRES activity regulation for increased p53 translation in response to etoposide.

The experiment in HEK293 cells would point towards a minor role of TIA1 in p53 translation control. The authors argue that this is because the experiment was not done in B cells. Therefore, they should do the effort to show that the situation is different if they would repeat the

experiment in B cells and that the p53 UTR is then indeed the translational control element that they argue it is.

Response – Very recently Jacob Stewart-Ornstein and Galit Lahav (*Science Signaling*, 25 Apr 2017: Vol. 10, Issue 476, eaah6671) have shown that p53 regulation varies in a cell-dependent manner. They evaluate the dynamic regulation of p53 in 12 cell lines in response to DNA damaging agents and conclude that differences in gene expression, cellular signalling and DNA repair efficiency are responsible for the very significant changes occurring in p53 regulation between cell lines. Thus, it is expected that p53 regulation is also substantially different in primary B cells.

Unfortunately, the technical challenges presented by primary mouse B cells do not facilitate the molecular analysis needed to support our hypothesis in a timely manner. Cell lines like HEK293 cells are widely used in cell biology as system to study the molecular mechanisms controlling gene expression and cell function. In order to provide further evidence about how RNA regulatory elements present in the p53 mRNA controls translation, we have performed the following:

1.- We generated new luciferase reporter constructs to functionally evaluate the role of the two U-rich sequences present in the p53 3'UTR that are bound by Tia1 and show that these function as regulatory elements (New figures 4A, 4C and 4H, and supplementary figures 5A and 5E. Discussed in page 9, lines 228-230). Site-directed mutagenesis of these sequences in p53 3'UTR increases mRNA translation of the luciferase reporter without changes in mRNA abundance, suggesting that they are indeed important for translational silencing.

2.- We used dsRNA to understand the role of endogenous Tia1 protein in p53 3'UTR-mediated mRNA translation (New figure 4B and supplementary figures 5B, 5C and 5D. Discussed in page 8, lines 222-227). We used three different pre-designed dsRNA and achieved different knock-down efficiencies (from 10-fold reduction to 1.2-fold). Tia1 knock-down consistently increased translation of a luciferase reporter mRNA containing the p53 3'UTR. Increased translation was independent of changes in mRNA abundance of the luciferase reporter mRNA and supports our previous conclusion that Tia1 expression silences p53 mRNA translation.

3.- We have evaluated whether overexpression of a wild-type Tia1 or of a Tia1-mutant isoform, lacking RNA recognition motifs 1 and 2 (Tia1 Δ RRM1+2), regulates p53 3'UTR-mediated mRNA translation (New figure 4D and supplementary figure 5F. Discussed in page 9, lines 232-233). Wild-type Tia1 overexpression consistently reduced translation of a luciferase reporter mRNA with a wild-type p53 3'UTR by 40%. Importantly, mutation of the two U-rich sequences bound by Tia1 rescued translation of the luciferase reporter. Furthermore, overexpression of a mutant Tia1 Δ RRM1+2 has no effect on mRNA translation of the luciferase reporter construct suggesting that direct binding of Tia1 to p53 3'UTR regulatory elements is required for mRNA translational silencing.

4.- We have generated two bicistronic mRNA luciferase reporter constructs to evaluate p53 5'UTR-mediated translation (New figure 6D, 6E and 6F; and supplementary figure 5H, 5I and 5J. Discussed in page 11, lines 292-303). Analysis of different ribosome footprinting datasets from mouse primary B cells, bone-marrow derived dendritic cells, mouse embryonic fibroblasts and mouse liver cells have all mapped a unique ribosome entry site in the p53 5'UTR. Therefore, we created a functional bicistronic mRNA luciferase reporter construct in which the ORF1 (encoding the renilla luciferase protein) was separated from the ORF2 (encoding the firefly luciferase protein) by the p53 5'UTR. Mutagenesis of the p53 IRES, defined by the ribosome footprinting data, reduces translation of a luciferase reporter construct by 5-fold. These new results reinforce our conclusion that p53 mRNA translation can be initiated in a CAP-independent manner. But, more importantly, p53 IRES-mediated translation was increased upon DNA damage induction with etoposide. This increase is completely blocked by mutation of the IRES sequence in the p53 5'UTR, suggesting that ribosome recognition of the p53 IRES is essential for p53 mRNA translation during the DNA damage response.

It is not very convincing to say that TIA1 inhibits translation, when the luciferase expression levels are higher in the full length than with the short version of the 3' UTR (figure 3L). That would more point towards activation of translation by the longer UTR than inhibition by TIA1. This issue needs much better experimental evidence.

Response – We accept the criticism of the reviewer and have therefore generated a more incisive set of reporters to analyse p53 regulation by Tia1. The full length p53 3'UTR contains RNA regulatory elements (e.g. the distal U-rich element close to the polyadenylation site that it is bound by Tia1) that are required for translational control. 3'UTR shortening might affect translation in ways that are yet uncharacterised. Therefore, we have removed those experiments in this new version of the manuscript and substituted them with a refined analysis of Tia1-mediated mRNA translation by site-directed mutagenesis of the Tia1 binding sites in the p53 3'UTR. In combination with our gain- and loss- of function experiments described above, we have added a significant amount of new experimental evidence that support strongly the role of Tia1 as a translational silencer of p53 expression.

Reviewers' Comments:

Reviewer #2:

Remarks to the Author:

Regulation of mRNA translation and subcellular location controls protein synthesis of key modulators of the DNA damage response during B cell activation

Diaz-Munoz MD et al (Turner M corresponding)

This is another revision of a manuscript that addresses the roles of stress granules and translational control in B lymphocyte activation and responses to DNA damage. The manuscript posits that in response to B cell activation that there is formation of RNA stress granules that feature TIA1 association with target gene transcripts, repressing their translation. DNA damage elicits dissociation of p53 mRNA from TIA1 protein in the RNA granules, enhancing translation of p53 mRNA. Enhanced translation of p53 is thought to occur via an IRES, with the subsequent elevated levels of p53 directing genes involved in the DNA damage response.

The manuscript addresses potentially significant ideas concerning p53 translation in lymphocyte development. Prior concerns involved the clarity of the experimental descriptions and the need to clarify the p53 translation control mechanism. The current revision of the manuscript addresses many of these concerns, while a few remain.

Reviewer concerns:

1. Figure 2: This reviewer found portions of the text describing the number and percentages of genes with changes in mRNA levels or translation difficult to follow (lines 148-178). For example in line 165: "By contrast, over 50% of the 3580 genes that showed changes in ribosome association showed significant changes in the amount of mRNA in at least one of these fractions (Fig. 2D)." By definition is not differential ribosome association defined as changes in the percentage of a given gene transcript in specific sucrose gradient fractions? Furthermore, many of the figures could be aided with descriptions of the abbreviations (e.g. LE/L in panel B and DE in panels C and D). In the Figure 2 legend, beginning at line 800: "mRNAs are grouped based on significant changes in abundance in any of the fractions (fractions 4 to 16) or in monosomes only (fractions 4 to 7), in light polysomes (fractions 8 to 10) or in heavy polysomes (fractions 9 to 16)." Fractions 9 and 10 represent both light and heavy polysomes?

2. The experiments in Figure 6 involving the ribosome mapping and bicistronic reporter are not sufficient to conclude that a cap-independent and IRES mechanisms underlie translational control of p53 in this model system. See for example work by Lloyd RE (RNA [2004] 10:720-730) for discussions on the limitations of the bicistronic reporter.

Reviewer #3:

Remarks to the Author:

The authors strengthened several aspects of their manuscript, which support the major part of their claims. Although I still think this may be a minor mechanisms contributing to p53 regulation, the results look robust enough to be believable.

There is one specific point about the translation inhibition in figure 6 that needs some attention. The authors use 4-EGI-1 as inhibitor of CAP-dependent translation. However, I miss a positive control. Therefore, the authors should include a control in which they show that their inhibitor worked in these cells.

Reviewers' comments:

Reviewer #2 (Remarks to the Author):

Regulation of mRNA translation and subcellular location controls protein synthesis of key modulators of the DNA damage response during B cell activation

Diaz-Munoz MD et al (Turner M corresponding)

This is another revision of a manuscript that addresses the roles of stress granules and translational control in B lymphocyte activation and responses to DNA damage. The manuscript posits that in response to B cell activation that there is formation of RNA stress granules that feature TIA1 association with target gene transcripts, repressing their translation. DNA damage elicits dissociation of p53 mRNA from TIA1 protein in the RNA granules, enhancing translation of p53 mRNA. Enhanced translation of p53 is thought to occur via an IRES, with the subsequent elevated levels of p53 directing genes involved in the DNA damage response.

The manuscript addresses potentially significant ideas concerning p53 translation in lymphocyte development. Prior concerns involved the clarity of the experimental descriptions and the need to clarify the p53 translation control mechanism. The current revision of the manuscript addresses many of these concerns, while a few remain.

Reviewer concerns:

1. Figure 2: This reviewer found portions of the text describing the number and percentages of genes with changes in mRNA levels or translation difficult to follow (lines 148-178). For example in line 165: "By contrast, over 50% of the 3580 genes that showed changes in ribosome association showed significant changes in the amount of mRNA in at least one of these fractions (Fig. 2D)." By definition is not differential ribosome association defined as changes in the percentage of a given gene transcript in specific sucrose gradient fractions?

Response – We have now revised the manuscript to bring more clarity to this section. Ribosome association quantifies the amount of a given transcript that is associated to ribosomes and separated in a sucrose gradient (independently of the fraction). On the contrary, polysome distribution analysis detects the changes in mRNA abundance in different fractions. This is now explained further at the beginning of this section.

Furthermore, many of the figures could be aided with descriptions of the abbreviations (e.g.LE/L in panel B and DE in panels C and D).

Response – Descriptions are added in figures and/or figure legends as suggested.

In the Figure 2 legend, beginning at line 800: “mRNAs are grouped based on significant changes in abundance in any of the fractions (fractions 4 to 16) or in monosomes only (fractions 4 to 7), in light polysomes (fractions 8 to 10) or in heavy polysomes (fractions 9 to 16).” Fractions 9 and 10 represent both light and heavy polysomes?

Response – We have now reviewed the text to correct the typo. Fractions 11 to 16 contain heavy polysomes.

2. The experiments in Figure 6 involving the ribosome mapping and bicistronic reporter are not sufficient to conclude that a cap-independent and IRES mechanisms underlie translational control of p53 in this model system. See for example work by Lloyd RE (RNA [2004] 10:720–730) for discussions on the limitations of the bicistronic reporter.

Response – Our novel approach using ribosome footprints has enabled us to map the p53 IRES at single nucleotide resolution. We have used bicistronic reporter constructs to validate the functionality of the p53 IRES. Our results are consistent with growing evidence from several labs indicating that the p53 IRES plays a major role in p53 translation in response to DNA damage (Cell. 2005 Oct 7;123(1):49-63.; Oncogene 2006, 25:4613-4619.; EMBO Rep 2006, 7:404-410.; 2013, Cell Death Differ 20:226-234.; Cell Cycle 2008, 7:2189-2198.; Oncogene 2013 33:611-618.). In these studies both DNA and RNA reporters were used and similar conclusions were reached. Lloyd et al. have reported that alternative splicing may occur when using bicistronic dual luciferase reporters. Cloning of a 5'UTR between the renilla luciferase and the firefly luciferase ORFs might introduce a splicing acceptor site resulting in deletion of the renilla luciferase coding sequence. This is 5'UTR sequence specific as not all 5'UTRs contain splicing acceptor sites. Quantitation of bicistronic dual luciferase mRNAs with two different qPCR assays designed to detect renilla luciferase and firefly luciferase ORFs can detect whether alternative splicing is occurring. In the absence of splicing the ratio renilla luciferase/firefly luciferase is equal to 1. On the contrary, this ratio is lower than 1 if the renilla luciferase ORF is deleted due to alternative splicing. Data presented in supplementary figure 5I shows that the ratio of renilla luciferase versus firefly luciferase in cells transfected with our reporter constructs is 1, as expected from a bicistronic reporter. On the contrary, the ratio is 0.5 for a psiCheck2 empty vector used as control. This is likely due to the presence of two different promoters that control DNA transcription of renilla luciferase and firefly luciferase independently. Thus, these data argue against alternative splicing affecting our reporters. This is now discussed in the manuscript (page 11).

Also, we have used different global inhibitors of translation and the CAP-mediated translation inhibitor 4EGI-1 to understand how p53 translation in B cells in response to etoposide. Our results are consistent with those obtained in MCF-7 cells and MEFs (Oncogene 2006, 25:4613-4619.; EMBO J. 2010 Jun 2;29(11):1865-76.). Taken together, we provided solid evidence indicating that regulation of translation, rather than transcription or protein stability, is responsible for p53 protein expression in B cells during the DNA damage response.

Reviewer #3 (Remarks to the Author):

The authors strengthened several aspects of their manuscript, which support the major part of their claims. Although I still think this may be a minor mechanisms contributing to p53 regulation, the results look robust enough to be believable.

There is one specific point about the translation inhibition in figure 6 that needs some attention. The authors use 4-EGI-1 as inhibitor of CAP-dependent translation. However, I miss a positive control. Therefore, the authors should include a control in which they show that their inhibitor worked in these cells.

Response – We thank the reviewer once again his/her careful consideration of our manuscript. We have now analysed the expression of cyclin D2 as a positive control of 4EGI-1- mediated inhibition of CAP-dependent translation (Figure 8B). Ccnd2 is a well-known target of eIF4E that is subjected to 4EGI-1 inhibition in several cell types (British Journal of Cancer (2012) 106, 1660–1667.). Ccnd2 expression is reduced by 3-fold in B cells after treatment with 4EGI-1 (Figure 8B).